# Reasoning on a Spectrum: Aligning LLMs to System 1 and System 2 Thinking

## Abstract

Large Language Models (LLMs) exhibit impressive reasoning abilities, yet their reliance on structured step-by-step processing reveals a critical limitation. In contrast, human cognition fluidly adapts between intuitive, heuristic (System 1) and analytical, deliberative (System 2) reasoning depending on the context. This difference between human cognitive flexibility and LLMs' reliance on a single reasoning style raises a critical question: while human fast heuristic reasoning evolved for its efficiency and adaptability, is a uniform reasoning approach truly optimal for LLMs, or does its inflexibility make them brittle and unreliable when faced with tasks demanding more agile, intuitive responses? To answer these questions, we explicitly align LLMs to these reasoning styles by curating a dataset with valid System 1 and System 2 answers, and evaluate their performance across reasoning benchmarks. Our results reveal an accuracy-efficiency trade-off: System 2-aligned models excel in arithmetic and symbolic reasoning, while System 1-aligned models perform better in commonsense reasoning tasks. To analyze the reasoning spectrum, we interpolated between the two extremes by varying the proportion of alignment data, which resulted in a monotonic change in accuracy. A mechanistic analysis of model responses shows that System 1 models employ more definitive outputs, whereas System 2 models demonstrate greater uncertainty. Building on these findings, we further combine System 1- and System 2-aligned models based on the entropy of their generations, without additional training, and obtain a dynamic model that outperforms across nearly all benchmarks. This work challenges the assumption that step-by-step reasoning is always optimal and highlights the need for adapting reasoning strategies based on task demands.

## 1 Introduction

LLMs have demonstrated remarkable reasoning capabilities, often achieving near-human or even superhuman performance (Huang & Chang, 2023). These advances have largely been driven by techniques that simulate step-by-step, deliberative reasoning, such as Chain-of-Thought (CoT) prompting and inference-time interventions (Wei et al., 2022b; Wang et al., 2022). Given their success, such methods are increasingly integrated into LLM training (Chung et al., 2024), reinforcing explicit, structured reasoning regardless of the task necessity. However, the increasing focus on step-by-step reasoning has revealed limitations such as brittle generalization, particularly in tasks requiring nuanced judgment (Delétang et al., 2023), logical consistency (Jiang et al., 2024), or adaptability to uncertainty (Mirzadeh et al., 2024). Similarly, recent analyses frame this issue as "overthinking" (Cuadron et al., 2025); Chen et al. (2024) demonstrate that excessive deliberation can hamper decision-making. This problem appears in LLMs' responses to simple factual queries, where they often generate unnecessarily long explanations instead of direct responses (Wang et al., 2023).

This focus on explicit, structured reasoning highlights a key difference between LLMs and human cognition: while LLMs are being pushed towards a single mode of processing, human reasoning is far more nuanced. Rather than a monolithic process, human reasoning emerges from a repertoire of cognitive tools evolved to tackle a *spectrum* of computational problems. This spectrum encompasses both automatic and reflective processes, a key insight recognized across diverse fields from economics to psychology and neuroscience (Daw et al., 2005; Dolan & Dayan, 2013; Balleine & Dickinson, 1998). On one end lie computationally *light* problems demanding rapid, intuitive judgments with

confidence (e.g., instinctively dodging a speeding car), handled by the reflexive "System 1 ($\mathcal{S}1$)." On the other end are *heavy* problems requiring deliberate, step-by-step analysis with prospection, managed by the reflective "System 2 ($\mathcal{S}2$)" (Kahneman, 2011; Stanovich & West, 2000). This dual-process system allows us to dynamically shift between modes depending on the task, balancing speed and accuracy (Evans & Stanovich, 2013). Extensive work in neuroscience over the past two decades links the dual-process framework and human decision strategies, which depicts decision-making on a spectrum between a fast but reflexive habitual decision strategy (Daw et al., 2011; Gillan et al., 2016; Miller et al., 2017) and a reflective goal-directed strategy (Daw et al., 2005; Dolan & Dayan, 2013). Experimental work in neuroscience is built on the relative advantages of these strategies, the separate but overlapping neural structures supporting them, and the circumstances under which each system is deployed in the brain (Schad et al., 2020; Piray & Daw, 2021). Given the evolutionary advantage of switching between fast and slow thinking to balance speed, efficiency, and accuracy, exploring LLMs through the lens of dual-process theory offers a powerful way to address their limitations.

While recent studies explore whether LLMs exhibit $\mathcal{S}1$ and $\mathcal{S}2$ behaviors (Hagendorff et al., 2023; Pan et al., 2024) or propose hybrid models (Yang et al., 2024; Deng et al., 2024), most prior work implicitly assumes that structured, deliberative reasoning is universally superior. Even research suggesting LLMs' capacity for both reasoning modes (Wang & Zhou, 2024) largely overlooks the crucial question of when each mode is indeed advantageous. The assumption that a single "best" reasoning strategy can apply across all contexts is a fundamental simplification that limits current approaches in LLM development. This assumption prevents LLMs from achieving human-like cognitive flexibility, hindering their ability to adapt their reasoning processes to diverse situations.

To address this gap, we design an experimental setup where both thinking styles can produce valid responses but follow distinct paths, one leveraging intuitive heuristics, and the other prioritizing deliberate, step-by-step reasoning. To implement this setup, we first curate a dataset of 2,000 reasoning questions where each problem has both $\mathcal{S}1$ and $\mathcal{S}2$ responses, grounded in ten well-studied cognitive heuristics (Tversky & Kahneman, 1974). Next, we explicitly align LLMs with either $\mathcal{S}1$ or $\mathcal{S}2$ responses and systematically assess them across diverse reasoning benchmarks. Our findings mirror the well-known accuracy–efficiency trade-off in human cognition (Keramati et al., 2011; Mattar & Daw, 2018): $\mathcal{S}2$–aligned models excel in arithmetic and symbolic reasoning, demonstrating superior multi-step inference but producing longer, token-intensive outputs, while $\mathcal{S}1$–aligned models generate succinct answers and perform better on commonsense reasoning tasks where heuristic shortcuts are effective. Beyond this trade-off, we also show that $\mathcal{S}1$ models are more confident and decisive, whereas $\mathcal{S}2$ models express greater uncertainty and hedge more, mirroring patterns observed in neuroscience (Daw et al., 2005). Then, to further examine this spectrum, we interpolated between the two extremes by varying the proportion of alignment data, which yielded a monotonic change in accuracy. Finally, we propose a training-free dynamic model that adaptively chooses between $\mathcal{S}1$ and $\mathcal{S}2$ reasoning based on output entropy signals. By framing LLM reasoning as a structured and adaptable process, this work highlights the importance of selecting the right reasoning strategy for a given task and sets the stage for more flexible, efficient, and robust reasoning systems. [1]

## 2 RELATED WORK

**Reasoning in LLMs.** Extensive research highlights both the strengths and weaknesses of LLM reasoning (Huang & Chang, 2022; Mondorf & Plank, 2024; Valmeekam et al., 2022; Parmar et al., 2024; Sourati et al., 2024; Shojaee et al., 2025). Recent efforts to enhance these abilities have largely focused on prompting (Brown et al., 2020), from zero-shot prompting with explicit instructions (Kojima et al., 2022; Wang et al., 2023; Zhou et al., 2024b) to few-shot prompting with step-by-step examples (Wei et al., 2022b). Wang & Zhou (2024) further show that CoT reasoning can be elicited from pre-trained LLMs by output decoding without a CoT prompt. Self-consistency decoding (Wang et al., 2022) improves robustness through diverse reasoning paths, aligning with $\mathcal{S}2$ reasoning. Tree of Thought (Yao et al., 2024) generalizes CoT, enabling LMs to explore multiple reasoning paths, self-evaluate, and look ahead or back to make global decisions. LLM reasoning can also be improved via CoT instruction tuning (Chung et al., 2024; Huang et al., 2022) or distillation (Magister et al., 2022), enabling models to internalize step-by-step reasoning and surpass prompting techniques.

---

[1] Our data and code are available at https://anonymous.4open.science/r/system12-004B

Figure 1: (A) Sample of dataset with System 1 and System 2 answers. (B) Overview of our alignment approach with fast and slow thinking. (C) Overview of our dynamic entropy based selection method.

Concurrent studies have identified an "overthinking" phenomenon in LLMs, where models generate excessively detailed or unnecessary reasoning steps (Chen et al., 2024; Cuadron et al., 2025).

**Dual-Process Theory in LLMs.** Dual-process theory offers a powerful framework for understanding human reasoning, though its use in NLP is still relatively underexplored. Existing research broadly falls into two main categories: First, researchers have investigated whether LLMs exhibit reasoning behaviors aligned with $\mathcal{S}1$ and $\mathcal{S}2$, particularly in terms of cognitive human-like errors and biases (Pan et al., 2024; Echterhoff et al., 2024; Zeng et al., 2024). Specifically, Hagendorff et al. (2023) examine cognitive heuristics in LLMs, showing that newer models exhibit fewer errors characterize with $\mathcal{S}1$ thinking. Booch et al. (2021) discuss fundamental questions regarding the role of dual-process theory in ML but leave practical implementation as an open problem. Second, several studies have integrated dual-process-inspired reasoning into LLMs. Some works combine intuitive (fast) and deliberate (slow) components to improve reasoning and planning (He et al., 2024; Liu et al., 2022; Hua & Zhang, 2022; Pan et al., 2024; Su et al., 2025; Saha et al., 2025), while others optimize efficiency by distilling $\mathcal{S}2$ insights into $\mathcal{S}1$ models (Yang et al., 2024; Deng et al., 2024; Yu et al., 2024). Research has also leveraged $\mathcal{S}2$ reasoning to mitigate biases associated with $\mathcal{S}1$ heuristics to improve fairness and robustness (Furniturewala et al., 2024; Kamruzzaman & Kim, 2024; Weston & Sukhbaatar, 2023). While most studies frame $\mathcal{S}2$ as superior and portray $\mathcal{S}1$ as erroneous despite its role in efficient reasoning, we instead investigate the implicit effects of aligning LLMs to either system. By analyzing how these heuristics shape general reasoning, we address a gap in the literature and offer new insights into broader cognitive behaviors of LLMs.

## 3 METHOD

### 3.1 ALIGNING LLMS TO SYSTEM 1 & SYSTEM 2 THINKING

We formalize fast and slow thinking as an alignment problem using a curated dataset where each reasoning question is paired with both $\mathcal{S}1$ (intuitive) and $\mathcal{S}2$ (analytical) responses (see Section 3.3). We align LLMs to either reasoning style via a preference-based training approach: for $\mathcal{S}1$ alignment, the intuitive response is designated as the preferred (winner) and the analytical response as the non-preferred (loser); for $\mathcal{S}2$ alignment, this preference is reversed, treating the analytical response as the winner and the intuitive response as the loser.

This approach is effective for two reasons: First, previous research has shown that prompt engineering can guide LLMs toward $\mathcal{S}2$ reasoning (Wei et al., 2022a) or $\mathcal{S}1$ reasoning (Zhou et al., 2024a), suggesting that LLMs already have both reasoning abilities. Therefore, instead of creating new reasoning pathways, we guide the model to distinguish between intuitive and analytical reasoning without altering its underlying knowledge. Given that both modes are already latent in pretrained LLMs, aligning the model to these two styles simply sharpens and separates capabilities that naturally coexist. Second, our aim is not to introduce new knowledge or instructions but rather to shape the model's reasoning process based on existing capabilities.

### 3.2 ENTROPY-BASED ARBITRATION BETWEEN REASONING STYLES

To create a dynamic model, we propose a training-free approach that arbitrates between $\mathcal{S}1$– and $\mathcal{S}2$-aligned models dynamically. The method adaptively selects the reasoning style best suited to

a given query using entropy-based signals. To quantify LLM uncertainty, we compute token-level entropy for each generated sequence of tokens $T = (t_1, \ldots, t_n)$ over vocabulary $V$:

$$H_i = -\sum_{v \in V} P(v|t_{<i}, x) \log P(v|t_{<i}, x), \tag{1}$$

where $P(v|t_{<i}, x)$ is the probability of token $v$ given the input $x$ and preceding tokens $t_{<i}$. From these token-level entropies, we calculate the average sequence entropy $\bar{H}$ and its variance $\sigma^2$:

$$\bar{H} = \frac{1}{n}\sum_{i=1}^{n} H_i, \quad \sigma^2 = \frac{1}{n}\sum_{t=1}^{n}(H_i - \bar{H})^2. \tag{2}$$

$\bar{H}$ captures the overall uncertainty of the model's predictions, while $\sigma^2$ reflects the instability of its reasoning process. "Stable and confident" predictions correspond to low values of both, "cautious but consistent" predictions arise from high $\bar{H}$ with low $\sigma^2$, and "instability" is signaled by high $\sigma^2$ regardless of $\bar{H}$. To enable comparison between $\mathcal{S}1$ and $\mathcal{S}2$ models, we denote their entropy statistics as $(\bar{H}_1, \sigma_1^2)$ and $(\bar{H}_2, \sigma_2^2)$, and normalize them via total sum scaling across the two systems, yielding $(\hat{H}_1, \hat{\sigma}_1^2)$ and $(\hat{H}_2, \hat{\sigma}_2^2)$. We then define the reliability $R_i$ for each model as a combined score:

$$R_i = w \times \hat{H}_i + (1-w) \times \hat{\sigma}_i^2, \quad 0 \leq w \leq 1. \tag{3}$$

For each question, the system with the lower score is selected. Recent works on reasoning stability (You et al., 2025; He et al., 2025; Ling et al., 2025) suggest penalizing instability more heavily than caution ($0 \leq w < \frac{1}{2}$). This scheme prioritizes "stable and confident" reasoning, accepts "cautious but consistent" reasoning, and penalizes "unstable" reasoning. In this way, the dynamic model outputs the most reliable answer between either $\mathcal{S}1$ or $\mathcal{S}2$ based on entropy signals without additional training.

### 3.3 Dataset of System 1 & System 2 Thinking

Our curated dataset consists of 2,000 questions, each paired with two responses that capture distinct reasoning styles in English: one intuitive and rapid, reflecting cognitive shortcuts ($\mathcal{S}1$), and the other deliberate and analytical ($\mathcal{S}2$). This dual structure provides a controlled setting to examine the mechanisms underlying $\mathcal{S}1$ and $\mathcal{S}2$ reasoning (Kahneman, 2011; Stanovich & West, 2000; Evans & Stanovich, 2013). The dataset was constructed in three phases:

**Generation.** To construct our dataset, we adopted a human-in-the-loop pipeline with GPT-4o (Hurst et al., 2024) to scale high-quality reasoning examples. In line with recent work on dataset creation using LLMs (Xu et al., 2023; Wang et al., 2022), we used a one-shot prompting setup, where each generation is guided by a seed example grounded in a cognitive heuristic, providing a practical foundation for distinguishing $\mathcal{S}1$ from $\mathcal{S}2$ reasoning (Kahneman, 2011). These seed examples, authored by domain experts (i.e., cognitive scientists; see Appendix E), cover 10 well-known heuristics from Kahneman (2011) (Appendix D). For each heuristic, experts provided a reasoning question with both a $\mathcal{S}1$ (heuristic) and $\mathcal{S}2$ (deliberative) response. During expansion, the prompt included the heuristic definition, descriptions of both systems approaches, and the expert-written example, enabling the model to generate new reasoning items aligned with distinct cognitive patterns. Early experiments showed that outcome-focused examples did not meaningfully guide model behavior. Thus, rather than mimicking naturalistic human responses, we designed process-oriented examples that explicitly articulate $\mathcal{S}1$ and $\mathcal{S}2$ reasoning. This helped models internalize distinct reasoning strategies beyond surface-level responses, as further supported in Sections 5 and 5.3 and appendix T . Prompt details and expert-authored examples are in Appendices F and G.

**Refinement.** As a byproduct of the data generation process, $\mathcal{S}2$ outputs were significantly longer and more detailed, reflecting step-by-step reasoning, while $\mathcal{S}1$ outputs were shorter and more direct (*Welch*'s test: $t(2090.1) = -184.74$, $p < .001$, $d = -5.84$). Prior work demonstrates that alignment methods can rely on superficial cues, such as output length, favoring longer responses even without reasoning advantages (Singhal et al., 2023). To prevent this bias, we use zero-shot prompting with GPT-4o to match the lengths of our $\mathcal{S}1$ and $\mathcal{S}2$ outputs while preserving content. Adjustments were applied only for significant length disparity. Details on the prompt and the length disparity threshold are in Appendix L. By reducing the length disparity, we minimized any preference for $\mathcal{S}2$ outputs

arising from their longer responses. After adjustment, $\mathcal{S}1$ outputs averaged 82.19 tokens, while $\mathcal{S}2$ outputs averaged 83.93 tokens. A two one-sided t-test (TOST) confirmed the equivalence of post-adjustment lengths across various token counts as equivalence margins (see Appendix K), indicating that the adjustment effectively eliminated significant length differences between the response types.

**Verification.**   Prior works show that high-quality, expert-supervised datasets of this scale are common and effective for LLM fine-tuning (Xiao et al., 2024; Dumpala et al., 2024; Li et al., 2024). Following this precedent to ensure data quality, we had our expert cognitive scientists conform all generated data to formal definitions of $\mathcal{S}1$ and $\mathcal{S}2$ thinking, and ensured that the dataset covers the intended set of cognitive heuristics across varied subjects. In this process, experts manually revised approximately 20% of the responses. We further verified the breadth of topic coverage via topic modeling; for more details and a sample of the curated dataset, see Appendices H and I.

## 4   EXPERIMENTS SETUP

**Alignment Algorithm.**   To implement the alignment strategy for $\mathcal{S}1$ and $\mathcal{S}2$ reasoning, we utilize two offline preference optimization methods, namely, Direct Preference Optimization (DPO; Rafailov et al., 2024) and Simple Preference Optimization (SimPO; Meng et al., 2024), for two reasons: (i) their offline formulation removes the costly on-policy sampling loop, yielding a simpler and more compute-efficient training pipeline, and (ii) our hand-crafted preference pairs capture fine-grained relational signals that would likely be blurred by online-generated pairs (more details in Appendix N).

**Benchmarks.**   We evaluate our models on 14 reasoning benchmarks across three different categories: (1) arithmetic reasoning: MultiArith (Roy & Roth, 2015), GSM8K (Cobbe et al., 2021), AddSub (Hosseini et al., 2014), AQUA-RAT (Ling et al., 2017), SingleEq (Koncel-Kedziorski et al., 2015), SVAMP (Patel et al., 2021), and AGIEval (Zhong et al., 2024); (2) commonsense reasoning: CSQA (Talmor et al., 2019), StrategyQA (Geva et al., 2021), PIQA (Bisk et al., 2020), SIQA (Sap et al., 2019), and COM2SENSE (Singh et al., 2021); (3) symbolic reasoning: Last Letter Concatenation and Coin Flip (Wei et al., 2022b). Following Kong et al. (2024), our evaluation follows a two-stage process. In the first stage, we present benchmark questions to model and record their responses. In the second stage, we prompt the model with the original question, its initial response, and benchmark-specific instructions to ensure the output is formatted as required. See Appendices J and O for benchmark details and instructions.

**Implementation Details.**   We use Llama-3.2-3B-Instruct, Llama-3.1-8B-Instruct, Llama-3.1-70B-Instruct (AI@Meta, 2024), and Mistral-7B-Instruct-v0.1 (Jiang et al., 2023) as SFT models for alignment. Following Kojima et al. (2023), we compare the performance of these aligned models against their instruction-tuned counterparts under zero-shot and zero-shot CoT prompting (details in Appendix P). To analyze the model's behavior along the $\mathcal{S}1$ to $\mathcal{S}2$ reasoning spectrum, we train seven intermediate models, where the winner responses are mixed at predefined ratios between $\mathcal{S}1$ and $\mathcal{S}2$. This structured interpolation allows us to systematically assess whether the transition between reasoning styles is discrete or gradual.

## 5   RESULTS

### 5.1   DISTINCT STRENGTHS OF SYSTEM 1 & SYSTEM 2 MODELS

Table 1 shows a comparison of exact matching accuracy across 14 benchmarks for Llama models at different scales (3B, 8B, and 70B). Specifically, we compare the base models with the dynamic models, $\mathcal{S}1$ and $\mathcal{S}2$ variants, and include results for CoT prompting for reference. Corresponding results for the Mistral model are available in appendix Q. Our findings reveal distinct performance trends for the $\mathcal{S}1$ and $\mathcal{S}2$ models, highlighting their respective strengths in different reasoning benchmarks.

In all arithmetic benchmarks (MultiArith, GSM8K, AddSub, AQuA, SingleEq, SVAMP, and AGIEval) with various difficulty, $\mathcal{S}2$ models consistently outperformed both the base models and their $\mathcal{S}1$ counterparts. This improvement is most significant in the AddSub and SingleEq benchmarks. Similarly, $\mathcal{S}2$ models outperformed $\mathcal{S}1$ models in nearly all symbolic reasoning benchmarks (Coin

Table 1: Accuracy comparison of our $\mathcal{S}1$, $\mathcal{S}2$, and Dynamic models based on Llama-3 models against instruction-tuned and CoT baselines across benchmarks. Each cell shows accuracy, with parentheses indicating the difference from the base model. Color intensity reflects the magnitude of deviation.

| | | Arithmetic | | | | | | | Symbolic | | Common Sense | | | | |
|---|---|---|---|---|---|---|---|---|---|---|---|---|---|---|---|
| | | MultiArith | GSM8K | AddSub | AQuA | SingleEq | SVAMP | AGIEval | Coin | Letter | CSQA | Strategy | PIQA | SIQA | COM2SENSE |
| System 2 | DPO | 98.99 (+0.78) | 96.74 (+2.06) | 89.68 (+3.75) | 51.06 (+0.19) | 94.83 (+3.51) | 86 (+2.3) | 47.2 (+1.4) | 94.8 (-0.2) | 90.2 (+2.0) | 69.62 (-3.94) | 61.39 (-7.67) | 85.46 (-0.31) | 72.06 (-2.83) | 76.42 (-4.07) |
| | SIMPO | 98.61 (+0.4) | 95.37 (+0.69) | 91.45 (+5.52) | 55.53 (+4.66) | 95.12 (+3.8) | 86 (+2.3) | 46.6 (+0.8) | 95 (0) | 88 (-0.2) | 73.49 (-0.07) | 67.73 (-1.33) | 83.94 (-1.83) | 73.68 (-1.21) | 79.99 (-0.5) |
| | Llama-70B | 98.21 | 94.68 | 85.93 | 50.87 | 91.32 | 83.7 | 45.8 | 95 | 88.2 | 73.56 | 69.06 | 85.77 | 74.89 | 80.49 |
| | Llama-70B-CoT | 98.39 | 94.74 | 86.18 | 50.91 | 91.77 | 84.3 | 45.6 | 96.2 | 88.4 | 72.92 | 69.59 | 85.77 | 75.08 | 79.85 |
| | Dynamic-DPO | 98.41 ↑ | 95.35 ↑ | 87.32 ↑ | 50.84 ↓ | 92.79 ↑ | 85.3 ↑ | 46.6 ↑ | 93.8 ↓ | 90 ↑ | 74.87 ↑ | 69.55 ↑ | 85.99 ↑ | 75.21 ↑ | 80.83 ↑ |
| | Dynamic-SIMPO | 98.57 ↑ | 95.19 ↑ | 89.43 ↑ | 53.21 ↑ | 92.06 ↑ | 84.3 ↑ | 46 ↑ | 94.2 ↓ | 87.8 ↓ | 73.69 ↑ | 69.47 ↑ | 85.52 ↓ | 74.93 ↑ | 81.33 ↑ |
| System 1 | DPO | 97.83 (-0.38) | 93.91 (-0.77) | 82.47 (-3.46) | 48.82 (-2.05) | 85.59 (-5.73) | 80.3 (-3.4) | 41.4 (-4.4) | 93.6 (-1.4) | 87.8 (-0.4) | 75.32 (+1.76) | 70.87 (+1.81) | 86.19 (+0.42) | 75.64 (+0.75) | 81.09 (+0.6) |
| | SIMPO | 97.5 (-0.71) | 94.28 (-0.4) | 81.94 (-3.99) | 49.69 (-1.18) | 90.23 (-1.09) | 83.1 (-0.6) | 44 (-1.8) | 92.8 (-2.2) | 87.6 (-0.6) | 73.87 (+0.31) | 69.62 (+0.56) | 85.85 (+0.08) | 75.32 (+0.43) | 81.46 (+0.97) |
| System 2 | DPO | 98.67 (+1.0) | 79.37 (+0.88) | 89.87 (+7.4) | 49.21 (+0.39) | 94.37 (+3.65) | 85.4 (+4.9) | 33 (+2.8) | 93.8 (-0.4) | 86.2 (+2.2) | 71.42 (0) | 60.87 (-6.68) | 81.15 (-2.01) | 67.93 (-3.19) | 76.42 (-2.6) |
| | SIMPO | 97.83 (+0.16) | 79.38 (+0.89) | 90.13 (+7.66) | 54.72 (+6.78) | 94.49 (+3.77) | 81.7 (+1.2) | 32.6 (+2.4) | 94.4 (+0.2) | 84.8 (+0.8) | 69.62 (-1.8) | 67.38 (-0.17) | 81.49 (-1.67) | 69.16 (-1.96) | 78.21 (-0.81) |
| | Llama-8B | 97.67 | 78.49 | 82.47 | 48.82 | 90.72 | 80.5 | 30.2 | 94.2 | 84 | 71.42 | 67.55 | 83.16 | 71.12 | 79.02 |
| | Llama-8B-CoT | 97.83 | 78.54 | 82.03 | 49.21 | 88.19 | 80.9 | 30.4 | 94.8 | 84.2 | 71.58 | 67.38 | 83.34 | 70.97 | 79.86 |
| | Dynamic-DPO | 98.87 ↑ | 79.15 ↑ | 88.07 ↑ | 48.93 ↑ | 93.62 ↑ | 84.80 ↑ | 31.80 ↑ | 93.80 ↓ | 86.00 ↑ | 71.96 ↑ | 69.78 ↑ | 83.74 ↑ | 72.16 ↑ | 79.34 ↑ |
| | Dynamic-SIMPO | 97.58 ↓ | 79.16 ↑ | 88.87 ↑ | 54.49 ↑ | 93.01 ↑ | 81.30 ↑ | 30.80 ↑ | 94.00 ↑ | 84.40 ↑ | 71.62 ↑ | 68.17 ↑ | 83.23 ↑ | 71.69 ↑ | 80.05 ↑ |
| System 1 | DPO | 98.5 (+0.83) | 77.01 (-1.48) | 80.76 (-1.71) | 46.46 (-2.36) | 77.24 (-13.48) | 78 (-2.5) | 27.8 (-0.4) | 93.4 (-0.8) | 83.8 (-0.2) | 72.81 (+1.39) | 68.21 (+0.66) | 83.94 (+0.78) | 72.16 (+1.04) | 79.99 (+0.97) |
| | SIMPO | 97.5 (-0.17) | 77.79 (-0.7) | 80.51 (-1.96) | 48.03 (-0.79) | 87.4 (-3.32) | 79.3 (-1.2) | 28.4 (-1.8) | 90 (-4.2) | 83.8 (-0.2) | 72.32 (+0.9) | 67.73 (+0.18) | 83.35 (+0.19) | 71.67 (+0.55) | 81.46 (+2.44) |
| System 2 | DPO | 75.88 (+0.56) | 69.73 (+0.97) | 74.45 (+3.97) | 32.01 (+1.5) | 67.44 (+1.1) | 54.3 (+0.4) | 27.2 (+2.6) | 76.8 (-1.2) | 66.4 (+0.2) | 59.27 (-3.12) | 67.95 (-1.11) | 71.23 (-1.92) | 44.96 (-0.86) | 58.34 (-1.63) |
| | SIMPO | 75.58 (+0.26) | 69.43 (+0.67) | 73.96 (+3.48) | 32.84 (+2.33) | 68.29 (+1.95) | 54.1 (+0.2) | 26.4 (+1.8) | 78 (0) | 66 (-0.2) | 59.97 (-2.42) | 66.78 (-2.28) | 72.42 (-0.73) | 44.71 (-1.11) | 58.89 (-1.08) |
| | Llama-3B | 75.32 | 68.76 | 70.48 | 30.51 | 66.34 | 53.9 | 24.6 | 78 | 66.2 | 62.39 | 69.06 | 73.15 | 45.82 | 59.97 |
| | Llama-3B-CoT | 75.32 | 69.08 | 71.97 | 31.24 | 66.59 | 53.7 | 25.2 | 78.2 | 66.4 | 62.14 | 69.26 | 73.44 | 45.63 | 59.97 |
| | Dynamic-DPO | 75.63 ↑ | 68.91 ↑ | 73.96 ↑ | 31.24 ↑ | 66.85 ↑ | 54.1 ↑ | 26.6 ↑ | 76 ↓ | 66.2 | 59.97 ↓ | 69.32 ↑ | 74.53 ↑ | 45.49 ↓ | 60.03 ↑ |
| | Dynamic-SIMPO | 75.41 ↑ | 68.96 ↑ | 72.28 ↑ | 31.24 ↑ | 67.44 ↑ | 53.9 | 26 ↑ | 77.6 ↓ | 65.8 ↓ | 61.73 ↓ | 69.51 ↑ | 73.79 ↑ | 46.06 ↑ | 59.27 ↓ |
| System 1 | DPO | 74.91 (-0.41) | 67.36 (-1.4) | 68.29 (-2.19) | 29.84 (-0.67) | 64.97 (-1.37) | 51.5 (-2.4) | 22 (-2.6) | 75.8 (-2.2) | 63.8 (-2.4) | 61.73 (-0.66) | 69.48 (+0.42) | 74.61 (+1.46) | 45.82 (0) | 60.08 (+0.11) |
| | SIMPO | 75.16 (-0.16) | 68.03 (-0.73) | 68.11 (-2.37) | 30.27 (-0.24) | 62.66 (-3.68) | 51.3 (-2.6) | 22.8 (-1.8) | 77.2 (-0.8) | 64.2 (-2.0) | 62.34 (-0.05) | 70.18 (+1.12) | 74.53 (+1.38) | 46.46 (+0.64) | 59.97 (0) |

and Letter), which require pattern recognition and logical structuring, further validating the idea that deliberative, slow-thinking models enhance performance in structured reasoning. While both approaches achieve high accuracy, $\mathcal{S}1$'s reliance on heuristic shortcuts introduce small but systematic errors that $\mathcal{S}2$'s deliberate, stepwise computations tend to avoid, such as rounding the number or adding numbers without checking. These findings are further supported by our AddSub analysis in Appendix T.

Conversely, $\mathcal{S}1$ models consistently excelled all of their $\mathcal{S}2$ counterparts, the base models, and the CoT variant on all commonsense reasoning benchmarks (CSQA, StrategyQA, PIQA, SIQA, and COM2SENSE), which depend on intuitive judgments and heuristic shortcuts. While $\mathcal{S}2$ reasoning is correct, its more deliberate nature can often lead to overthinking, producing overly cautious or extensively interpretive responses that diverge from typical human reactions in rapid, intuitive situations. For example, when asked what a kindergarten teacher does before nap time, $\mathcal{S}2$ suggests "encourage quiet behavior" instead of "tell a story," or predicts "laughter" rather than "fight" if you surprise an angry person. As shown in Appendix T, this tendency to favor completeness over contextual fit makes $\mathcal{S}2$ less reliable for quick, socially grounded tasks.

Llama models generally outperformed Mistral (See Appendix Q) across all benchmarks, suggesting stronger foundational reasoning capabilities further enhanced by $\mathcal{S}1$ and $\mathcal{S}2$ alignment. Moreover, instruction-tuned models with CoT prompts exhibited marginal gains over their base counterparts, as step-by-step reasoning is already internalized during pretraining on CoT data (AI@Meta, 2024). Accordingly, we adopt the base Llama 8B model as our primary baseline in subsequent experiments since it offers a good trade-off between resource efficiency and performance.

In summary, our results showcase that $\mathcal{S}2$ models excel in structured, multi-step reasoning such as arithmetic and symbolic reasoning, while $\mathcal{S}1$ models are effective in intuitive and commonsense reasoning benchmarks. These findings highlight the significant potential of dual-process alignment for boosting LLM performance across a diverse range of reasoning paradigms.

## 5.2 LENGTH DIFFERENCES ACROSS REASONING STYLES

A recent trend in LLM performance, exemplified by models such as DeepSeek R1 (Guo et al., 2025), is that achieving stronger benchmark results often correlates with producing longer reasoning chains, even if not explicitly trained to do so. This correlation raises the question of whether such verbose

responses truly reflect enhanced reasoning capabilities or if they are simply a formatting artifact of current high-performing models. In our studies, this concern is particularly relevant for $\mathcal{S}2$ models, which are expected to behave more deliberatively. To investigate this, we analyze output lengths with the two-stage prompting setup described in Section 4.

As shown in Figure 2, $\mathcal{S}2$-aligned models generate significantly longer responses than their $\mathcal{S}1$ counterparts, relative to the Llama baseline, under both alignment methods, DPO ($t(8836) = 58.978$, $p < .001$) and SimPO ($t(8586) = 11.24$, $p < .001$). This difference emerges specifically in the second stage, where models are prompted to finalize their responses, while response lengths remain comparable in the first stage, where both models are simply instructed to reason. Although both models were trained on equal-length preference pairs as described in Section 3.3, $\mathcal{S}2$ models still tend to elaborate more during finalization, consistent with their alignment toward deliberative reasoning.

While longer reasoning chains are often associated with stronger performance, our findings suggest that this extended reasoning can also introduce inefficiencies or even degrade quality in contexts where concise, heuristic-driven reasoning is more appropriate. In particular, tasks requiring commonsense or intuitive judgments are often better handled by $\mathcal{S}1$ models, which respond more directly. This aligns with emerging work on "overthinking" phenomenon, where excessive deliberation hurts performance (Chen et al., 2024; Cuadron et al., 2025). To confirm that "overthinking" behavior is an inherent property of the reasoning style, we conducted an ablation study with un-normalized

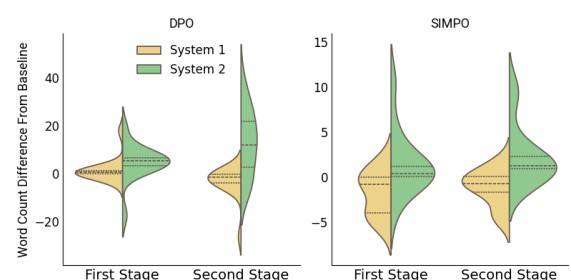

Figure 2: Token difference between System 1 and System 2 responses relative to Llama model across prompting stages and alignment methods.

data (See Appendix M). Overall, extended reasoning is not universally beneficial, and reasoning strategies must be evaluated in relation to the task.

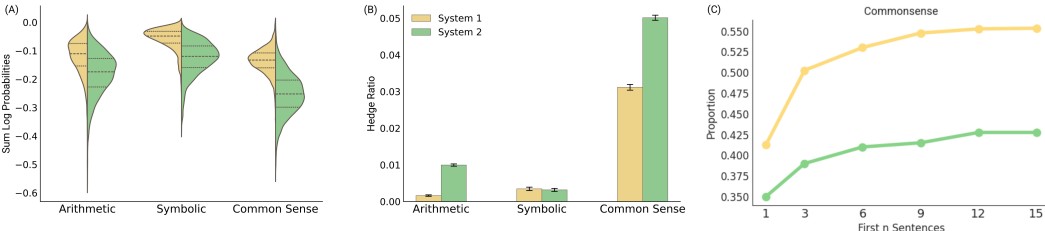

Figure 3: (A) Log probabilities of models' reasoning indicating internal uncertainty; (B) Hedge word ratio showing surface-level uncertainty; (C) Proportion of definitive answers in the first n sentences.

### 5.3 UNCERTAINTY ACROSS REASONING STYLES

A key insight from psychology and neuroscience is that $\mathcal{S}1$ operates on confident heuristics, providing quick, intuitive judgments, while $\mathcal{S}2$ engages in more deliberate, analytical thought, accurately assessing the uncertainty associated with its conclusions (Daw et al., 2005; Lee et al., 2014; Keramati et al., 2011; Xu, 2021). To examine uncertainty and confidence, we consider three different characteristics: 1) token-level uncertainty; 2) the presence of hedge words in model output (Lakoff, 1973; Ott, 2018); and 3) definitive commitment to responses in $\mathcal{S}1$ versus $\mathcal{S}2$.

Plot A in Figure 3 shows that $\mathcal{S}2$ models consistently generate tokens with lower confidence than $\mathcal{S}1$ models, based on token-level uncertainty from logits. This trend holds across arithmetic $t(4075) = 55.68, p < .001$, symbolic $t(999) = 42.53, p < .001$, and commonsense $t(3510) = 106.86, p < .001$ benchmarks. Additionally, we analyzed surface-level uncertainty in model reasoning by examining word choices. Figure 3, Plot B shows $\mathcal{S}2$-aligned models use significantly more hedge words, in arithmetic $t(4075) = 24.61, p < .001$ and commonsense $t(3510) = 21.49, p < .001$ when models reiterate their reasoning. While increased uncertainty

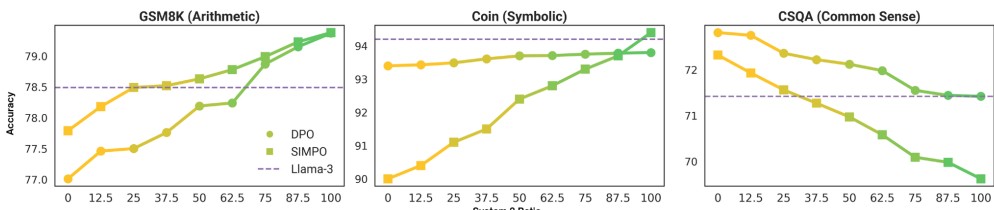

Figure 4: Accuracy across benchmark categories as reasoning shifts from System 1 to System 2.

enhances analytical reasoning, it may hinder tasks requiring rapid, intuitive judgments. To assess early-stage response conclusiveness, we used LLM-as-Judge (Zheng et al., 2023) as detailed in Appendix S. Figure 3, Plot C shows $\mathcal{S}1$ models provide significantly more definitive responses than $\mathcal{S}2$ models in commonsense reasoning, *McNemar's* $\chi^2(1, 400) = 20.0, p < .001$, regardless of where in the response the definitive responses are reached (see Appendix S).

This analysis reinforces the idea that different reasoning styles are suited to different tasks. Greater uncertainty in models' generated reasoning suggests that $\mathcal{S}2$ models can explore alternative reasoning paths more effectively. This uncertainty is reflected in both their model output probabilities and word choices. $\mathcal{S}2$ models' superior performance in arithmetic benchmarks highlights the benefits of deliberate, effortful processing in tasks that demand exploration and uncertainty. On the other hand, the greater tendency of $\mathcal{S}1$ models to commit to responses in a more definitive way aligns with their advantage in tasks requiring rapid and intuitive judgments. This behavior is observed exclusively in commonsense reasoning, where quick, decisive responses are advantageous—a trend supported by human studies (Byrd, 2022) and confirmed by our findings in Section 5.1. However, it does not appear in other benchmarks (see Appendix S), suggesting that the activation of a particular reasoning style is context-dependent and influenced by task demands.

## 5.4 MOVING FROM FAST TO SLOW THINKING

In the previous analysis, $\mathcal{S}1$ and $\mathcal{S}2$ models can be viewed as endpoints of a broader spectrum of reasoning strategies. Paralleling approaches in cognitive psychology (Daw et al., 2011; Piray & Daw, 2021), we explored this spectrum by constructing interpolated models—blending $\mathcal{S}1$ and $\mathcal{S}2$ preferred answers at varying ratios in the alignment dataset. Figure 4 demonstrates a consistent, monotonic transition in accuracy across representative benchmarks from three reasoning categories (all $r^2 > 0.9, p < 0.001$), a pattern visible across all benchmarks (see Appendix R). While arithmetic and symbolic reasoning benchmarks exhibit a steady increase in accuracy moving toward $\mathcal{S}2$ thinking, commonsense reasoning benchmarks show the opposite trend, with accuracy increasing as models rely more on $\mathcal{S}1$ reasoning. This trade-off highlights that both reasoning styles offer unique advantages, with $\mathcal{S}2$ excelling in structured, multi-step problem-solving and $\mathcal{S}1$ providing efficient, adaptable responses in intuitive scenarios. These findings strengthen the importance of task-dependent reasoning strategies that leverage the strengths of both $\mathcal{S}1$ and $\mathcal{S}2$ thinking. Critically, there are no sudden drops or fluctuations in performance when transitioning between reasoning styles. This stability indicates that the shift from $\mathcal{S}1$ to $\mathcal{S}2$ reasoning is gradual and predictable, without any unexpected anomalies. This observation reinforces the idea that LLMs can be strategically guided toward different reasoning styles, allowing for more adaptive problem-solving.

## 5.5 ENTROPY-GUIDED MODEL SELECTION

We evaluated the dynamic model proposed in Section 3.2 on our 14 reasoning benchmarks, varying the weight $w$ in Equation (3). As shown in Table 1 and Table 6, overall, the dynamic models consistently outperform their base counterparts across the different alignment algorithms on nearly all benchmarks. The best performance was achieved with $w = 0.4$, under which the Llama DPO-dynamic model achieved higher accuracy than the base model on 13 of the 14 benchmarks, while the SimPO-dynamic version improved on 12 benchmarks. Given the significance of this finding, we also replicated the analysis with Mistral models, where the DPO-dynamic model outperformed the base on 12 of 14 benchmarks, while the SimPO-dynamic model improved on 13 of 14 benchmarks (see Appendix U).

Furthermore, to validate the balance between uncertainty ($\hat{H}$) and instability ($\hat{\sigma}^2$) in our dynamic model, we analyzed the distributions of $\bar{H}$ and $\sigma^2$ between the two systems. As an illustration, Figure 5 shows GSM8k accuracy across different $w$ values alongside the corresponding entropy statistics; results for the remaining benchmarks are provided in Appendix U and follow the same trend. This analysis reveals systematic differences between correct and incorrect responses in $\mathcal{S}1$ and $\mathcal{S}2$ models. In general, high $\bar{H}$ in either system is associated with incorrect responses, whereas for both correct and incorrect cases the two systems exhibit very similar entropy statistics. We also observe that $\hat{H}$ is generally lower for $\mathcal{S}1$ models, indicating greater confidence, while $\hat{\sigma}^2$ is lower for $\mathcal{S}2$ models, indicating greater stability. These findings are consistent with Section 5.3 and with prior research in psychology and neuroscience. Together, this analysis provides empirical justification for using entropy signals as the basis of our scoring method in Section 3.2.

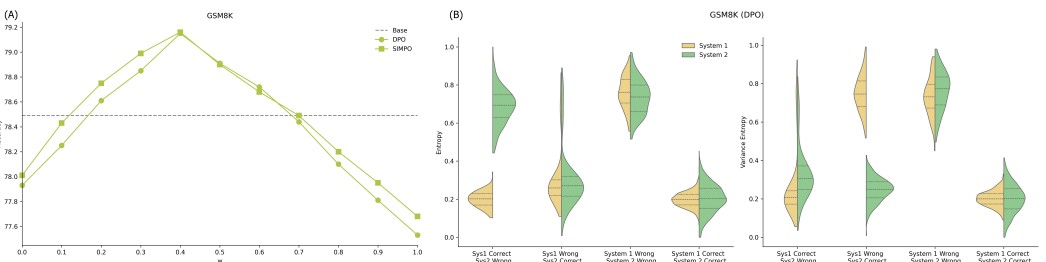

Figure 5: (A) Performance of Llama models (DPO- and SimPO-dynamic models) on the GSM8K dataset as $w$ varies in Equation (3). The dashed line represents the accuracy of the base Llama model. (B) Violin plots of average entropy ($\bar{H}$) and its variance ($\sigma^2$) distribution for DPO-aligned Llama models on GSM8K, broken down by four possible outcomes.

## 6 CONCLUSION

A central question in current LLM development is whether structured, step-by-step reasoning is always beneficial, or whether a more flexible range of reasoning strategies is needed. Inspired by dual-process theories of human cognition, we studied LLMs explicitly aligned with $\mathcal{S}1$ and $\mathcal{S}2$ thinking, representing fast, confident, heuristic reasoning and slow, analytical reasoning, respectively. Our findings indicate that, much like in human cognition, reasoning in LLMs is not a one-size-fits-all solution: different reasoning modes are effective in different contexts and downstream tasks. $\mathcal{S}2$ excels in arithmetic and symbolic reasoning, while $\mathcal{S}1$ is more effective and accurate in commonsense reasoning (Section 5.1). Additionally, $\mathcal{S}1$ models generate responses with fewer tokens, highlighting its efficiency in decision-making (Section 5.2). Our analysis in Section 5.3 illustrated that $\mathcal{S}2$ models exhibit greater uncertainty throughout the reasoning process, potentially resulting them to engage in more structured, step-by-step problem-solving. In contrast, $\mathcal{S}1$ models display higher confidence, allowing them to reach responses faster, which is particularly advantageous for tasks requiring rapid, intuitive judgments. Moreover, training intermediate models with blended ratios of preferred $\mathcal{S}1$ and $\mathcal{S}2$ responses revealed smooth, monotonic shifts in performance across benchmarks (Section 5.4), supporting the view that LLM reasoning should lie on a continuous, tunable spectrum rather than a binary divide. Finally, we proposed a dynamic model that selects adaptively between $\mathcal{S}1$ and $\mathcal{S}2$ reasoning based on entropy signals. Remarkably, this method requires no additional training yet consistently improves performance across diverse reasoning benchmarks. This demonstrates that our ensemble approach, guided by the model's confidence and the stability of that confidence to decide whether to rely on the System 1 or System 2 answer, can consistently produce the most reliable output. The method is adaptive in how it selects between the two reasoning modes, though it currently doubles the inference cost. An important direction for future work is to distill both reasoning modes into a single, efficient model to mitigate this overhead.

Beyond these empirical findings, our study aligns with broader principles observed across cognitive science and neuroscience. The observation that $\mathcal{S}1$ models generate faster and more confident responses echoes established theories in human cognition, where intuitive, heuristic thinking allows for rapid decision-making. Similarly, the higher uncertainty exhibited by $\mathcal{S}2$ models aligns with neuroscience findings that deliberate reasoning involves greater cognitive load, self-monitoring, and

exploring more paths. These parallels suggest that LLMs, when properly aligned, can mirror key aspects of human cognition, offering new insights into both artificial and natural intelligence.

Our work bridges between LLM development and cognitive science, highlighting efficiency-accuracy trade-offs in LLMs, similar to those long observed in human cognition. We align models with reasoning behaviors that follow well-known cognitive heuristics, which humans use in everyday thinking, like $\mathcal{S}1$'s rapid, intuitive judgments and $\mathcal{S}2$'s deliberate, analytical thought, and show they can follow the dynamic interplay between fast and slow thinking. This alignment not only informs more sophisticated training and evaluation strategies but also suggests that future LLMs can be designed to possess a more cognitively grounded flexibility, allowing them to adapt their reasoning as effectively as humans do when faced with diverse task demands. Finally, models that reason in ways that are cognitively interpretable, mirroring the human brain's strategies for learning, decision making, and inference, may also be more predictable, steerable, and trustworthy in deployment. In this light, dual-process alignment connects cognitive science and neuroscience with model capabilities, enabling future LLMs to reason more like humans, not just in what they conclude, but in how they get there.

This paper takes a first step toward adaptive reasoning in LLMs, enabling dynamic shift between heuristic and deliberative thinking based on task demands. Furthermore, understanding how to optimally balance speed and accuracy in LLMs can have significant implications for real-world applications, from conversational agents to automated decision-making systems. In practice, this approach allows deliberate trade-offs between answer quality and response speed, using fewer reasoning steps when time is critical.

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

## A  LIMITATIONS

Despite the promising advancements of using different thinking styles through the lens of dual-process cognitive theory in our approach, it is important to clarify the intended scope and outline future directions. Our curated dataset of 2,000 questions covers 10 well-established cognitive heuristics and was validated by our domain experts to ensure quality. While not exhaustive, this dataset provides a strong foundation for investigating reasoning style differences and establishes methodological groundwork for broader-scale expansion in future studies to represent the entire

spectrum of reasoning challenges encountered in real-world tasks. We focused our alignment experiments on Llama and Mistral as base models, using DPO and SIMPO as preference optimization techniques. While our findings are likely to generalize across model architectures and alignment methods, given the shared emergence of both intuitive and deliberative reasoning in large-scale pretraining, testing this generalization to other architectures and alignment methods is a valuable future direction. Moreover, while our dynamic model is training-free and improves performance, it is computationally inefficient. It doubles inference costs and memory usage by requiring both models to run for every query. Future work could distill this capability into a single, efficient model to mitigate this overhead. In terms of evaluating reasoning uncertainty, we adopt token-level logit-based measures and linguistic hedging analysis as computationally tractable proxies. These provide interpretable signals of reasoning behavior, though deeper psycholinguistic and interactive evaluations may offer complementary insights. Finally, while our experiments reveal a clear accuracy-efficiency trade-off between intuitive and deliberative reasoning, the extent to which these findings translate to more complex or efficient dynamic decision-making scenarios remains an open question. Future work should explore larger, more diverse datasets and investigate alternative alignment strategies to further validate and extend these results.

## B  ETHICAL STATEMENT

Aligning LLMs with $S1$ and $S2$ reasoning raises concerns about model behavior in different contexts. On one hand, $S1$ models risk producing overly confident but incorrect or biased responses, and their alignment with heuristics could be misinterpreted as an endorsement of harmful stereotypes. We want to be clear that the goal of this work is to leverage heuristics for their efficiency, not to amplify unfair biases. On the other hand, $S2$ models, though more deliberate, are not a universal solution as they introduce slower response times and increased computational costs. Responsible deployment requires building systems that engage the appropriate reasoning style for the context and strike a balance between efficiency and the risk of biased or misleading outputs.

## C  LLM USAGE

We used Large Language Models (specifically OpenAI's GPT models) exclusively for polishing the writing of this paper. No aspects of the research design, implementation, or analysis involved LLM assistance.

## D  COGNITIVE HEURISTICS

In Table 2, we list 10 different cognitive heuristics and their definitions, which we used in curating the dataset Kahneman (2011); Stanovich & West (2000); Evans & Stanovich (2013).

## E  DETAILS OF EXPERTS

The experts consulted are the three authors of this paper: two are PhD students and the other is a faculty member, all specializing in cognitive sciences.

## F  INITIAL DATA EXAMPLES

The 10 samples generated by the expert for our data generation are shown in Table 3.

## G  PROMPT FOR DATA EXPANSION

We expand our sample dataset by concatenating the expert-generated samples with the definitions in Table 2, along with a description of how $S1$ and $S2$ would respond to a given question, as shown below:

Table 2: 10 common cognitive biases and their definitions, which were considered in curating the dataset

| Cognitive Bias | Definition |
|---|---|
| Anchoring Bias | The tendency to rely too heavily on the first piece of information we receive about a topic, using it as a reference point for future judgments and decisions, even when new information becomes available. |
| Halo Effect Bias | The tendency to let one positive impressions of people, brands, and products in one area positively influence our feelings in another area. |
| Overconfidence Bias | The tendency to have excessive confidence in one's own abilities or knowledge. |
| Optimism Bias | The tendency to overestimate the likelihood of positive outcomes and underestimate negative ones. |
| Availability Heuristic Bias | The tendency to use information that comes to mind quickly and easily when making decisions about the future. |
| Status Quo Bias | The preference for maintaining the current state of affairs, leading to resistance to change. |
| Recency Bias | The tendency to better remember and recall information presented to us most recently, compared to information we encountered earlier |
| Confirmation Bias | The tendency to notice, focus on, and give greater credence to evidence that fits with our existing beliefs. |
| Planning Fallacy | The tendency to underestimate the amount of time it will take to complete a task, as well as the costs and risks associated with that task even if it contradicts our experiences. |
| Bandwagon Effect Bias | The tendency to adopt beliefs or behaviors because many others do. |

Table 3: 10 samples generated by an expert

| Category | Question | System 1 Answer | System 2 Answer |
|---|---|---|---|
| Anchoring Bias | Do you rely on your first impression of meeting your lab mate ? | Yes, my gut instinct is usually right. | I should interact with them more to form a well-rounded opinion. |
| Halo effect Bias | How do you feel about the new political candidate? | I do not like their stance on one issue, so I think they are a terrible candidate. | I'll weigh their stance on multiple issues before deciding. |
| Over Confidence Bias | Do you think you will succeed in your new job? | I will definitely succeed here. | I will need to put in effort and adapt to the new environment to succeed. |
| Status Quo Bias | Should you change your workout routine? | My routine has always worked, so there is no need to change it. | My fitness needs might have changed, so I will consider adjusting my routine. |
| Optimism Bias | Do you need to double-check your work after a mistake? | I am usually careful, so one mistake doesn't mean I'll make another. | I will double-check my work to make sure I don't repeat the mistake. |
| Availability heuristic | Is the newest seafood restaurant the best restaurant in town? | It is the most popular one, so it must be the best. | Popularity does not always mean the best quality, so I will read reviews first. |
| Recency Bias | Should you invest in the stock after hearing good things about it? | Yes, it is been rising lately, so it's sure to keep going up. | I will research the stock and market conditions before making a decision. |
| Confirmation Bias | Is the newest seafood restaurant the best restaurant in town? | It is the most popular one, so it must be the best. | Popularity does not always mean the best quality, so I will read reviews first. |
| Planning Fallacy | Is the newest seafood restaurant the best restaurant in town? | It is the most popular one, so it must be the best. | Popularity does not always mean the best quality, so I will read reviews first. |
| Bandwagon Effect Bias | Why did you pick apple as brand of your phone? | Everyone I know has this brand, so it must be the best. | I compared different features and chose the one that suits my needs. |

Table 4: Samples of curated dataset with the cognitive heuristic category, each paired with System 1 response using the heuristic and System 2 response mitigating that heuristic.

| Category | Question | System 1 Answer | System 2 Answer |
|---|---|---|---|
| Anchoring | When booking travel, do you find it hard to move away from the first deal you see? | I often find myself comparing everything to that initial deal. Once I've encountered a specific offer or price, it becomes the standard by which I measure all others. Even when new deals come along, my mind instinctively revisits that first one, guiding my judgment. It's an automatic comparison that influences how I evaluate options moving forward. | To book travel effectively, I avoid comparisons only against the initial option. I understand that the first deal isn't always the best ensures a broader perspective, more observations increases the chances of finding a more favorable travel plan. Exploring more options ultimately leads to better decision-making. |
| Halo Effect | Would you say fasting diet is effective? | This diet is obviously effective. just look at the people who follow it! Celebrities and influencers swear by it, and they're in great shape. When so many successful, healthy people use it, that's proof it works. Plus, it's super popular, which wouldn't happen if it didn't give great results. If you want to see real change, this is the way to go! | The effectiveness of a diet depends on scientific evidence, individual health factors, and long-term sustainability rather than its popularity or association with celebrities. A diet may lead to short-term weight loss but could lack essential nutrients or be difficult to maintain. Evaluating its impact on metabolism, overall health, and adherence is crucial.. |
| Over Confidence | Do you believe you will ace the interview? | Yes, I excel in interviews. I communicate clearly, stay confident under pressure, and listen attentively to questions. My ability to understand the interviewer's needs and align my responses accordingly enhances my effectiveness. I maintain engaging body language and make genuine connections, making a lasting impression. I prepare thoroughly, anticipate potential questions, and rehearse answers, ensuring I approach interviews with a calm, composed demeanor, making me a strong candidate. | While confidence in interpersonal skills is beneficial, thorough interview preparation is essential for success. It allows for anticipation of potential questions and crafting informed responses, showcasing an understanding of the company and role. Researching company culture enables candidates to align their answers with organizational values, enhancing their chances of making a positive impression. Solely relying on confidence can lead to unpreparedness, especially for technical inquiries, reducing the effectiveness of skill articulation. |

Figure 6: Topic modeling results on our dataset. Each dot represents a question, and colors indicate distinct topics.

```
The System 1 response should demonstrate intuitive, fast reasoning
that relies on the heuristic, showing the shortcut-like process it
uses.  The responses should highlight the reasoning style itself,
not just the final answer.
```

```
The System 2 response should demonstrate slow, step-by-step
reasoning that carefully analyzes the question, explicitly
contrasting with System 1.  The responses should highlight the
reasoning style itself, not just the final answer.
```

## H  FINAL DATASET SAMPLE

A subset of the curated dataset is shown in Table 4.

## I  TOPIC MODELING

Following expert validation, we experimentally verified the diversity of our dataset to ensure it goes beyond surface-level variation in wording. Figure 6 presents the results of topic modeling using BERTopic (Grootendorst, 2022), demonstrating the range of topics covered in the dataset. The wide distribution and clustering across 150 unique topics demonstrate the semantic diversity of the dataset beyond superficial lexical variation.

## J   BENCHMARK DETAILS

We use three categories of reasoning benchmarks: arithmetic, commonsense reasoning, symbolic reasoning, We provide an overview of the datasets used in each category.

**Arithmetic reasoning.**   We use seven datasets: MultiArith, GSM8K, AddSub, AQuA, SingleEq, SVAMP, and AGIEval. Each dataset consists of questions that present a scenario requiring numerical computation and multi-step reasoning based on mathematical principles.

**Commonsense reasoning.**   To assess commonsense reasoning, we utilize five benchmarks: CommonsenseQA (CSQA), StrategyQA, PIQA, SocialIQA (SIQA), and Com2Sense. All require models to go beyond surface-level understanding and reason using prior knowledge. CSQA focuses on multiple-choice questions grounded in general world knowledge, while StrategyQA includes questions that demand implicit multi-hop reasoning. PIQA evaluates physical commonsense by requiring models to choose the more plausible solution to everyday benchmarks. SIQA targets social commonsense, presenting scenarios about interpersonal interactions and asking questions about motivations, reactions, and emotions. Com2Sense provides pairs of complementary sentences to test a model's ability to distinguish between plausible and implausible statements using commonsense.

**Symbolic reasoning.**   We use the Last Letter Concatenation and Coin Flip datasets. Last Letter Concatenation involves forming a word by extracting the last letter of given words in order. Coin Flip presents a sequence of coin-flipping instructions and asks for the final coin orientation. These datasets were originally proposed by Wei et al. (2023a) but were not publicly available. Kojima et al. (2023) later followed their approach to create and release accessible versions, which we use in our experiments.

## K   EQUIVALENCE TESTING OF DATASET LENGTHS USING TOST

A two one-sided t-test (TOST) confirmed the equivalence of these post-adjustment lengths across various token counts as equivalence margins: $\pm 3$ tokens, $t(3870.30) = 85.82$, $p < .001$; $\pm 5$ tokens, $t(3870.30) = 149.07$, $p < .001$; $\pm 7$ tokens, $t(3870.30) = 212.31$, $p < .001$; and 5% of the mean token count ($\pm 4.15$ tokens), $t(3870.30) = 122.29$, $p < .001$

## L   LENGTH ADJUSTMENT THRESHOLD AND PROMPT

We adjust the length if there is a disparity of more than 15 tokens between the $\mathcal{S}1$ and $\mathcal{S}2$ outputs using GPT-4o with the following prompt:

```
For a given {question}, we have two types of answers:  A fast,
intuitive response based on cognitive heuristics which is our
System 1 Answer.
System 1 Answer:  {System 1 Answer}
And a slow, deliberate, and logical reasoning response which is our
System 2 Answer.
System 2 Answer:  {System 2 Answer}
Your task is to adjust the two answers so that they are presented
in the same order of tokens without altering their content.  Ensure
that the intuitive nature of the System 1 Answer and the logical
reasoning of the System 2 Answer are preserved.
```

## M   ABLATION STUDY ON LENGTH NORMALIZATION

We conducted an ablation study by training models on the un-normalized dataset, where $\mathcal{S}2$ responses were naturally longer than $\mathcal{S}1$ responses. We then analyzed the length of the responses generated by these models at inference time. $\mathcal{S}2$-aligned models generate significantly longer responses than their $\mathcal{S}1$ counterparts, relative to the Llama baseline, under both alignment methods, DPO

$(t(8836) = 71.831, p < .001)$ and SimPO $(t(8586) = 15.227, p < .001)$. This suggest that the "overthinking" behavior is inherent to the $\mathcal{S}2$ reasoning style, in both of the settings, the $\mathcal{S}2$ models generate more tokens compared to their $\mathcal{S}1$ counterparts.

## N    ALIGNMENT ALGORITHM

DPO is an offline alignment method that fine-tunes LLMs by comparing the preferred and disfavored outputs of a model against a reference model, optimizing preferences without requiring a separate reward model. As a prominent method in preference optimization, DPO has gained traction for its stability and efficiency, making it a widely adopted alternative to Reinforcement Learning from Human Feedback (RLHF; Ouyang et al., 2022). SimPO builds on the principles of DPO but introduces a reference-free approach to preference optimization. Instead of requiring a separate reference model, SimPO aligns responses by directly optimizing preference signals within the model itself. This makes it computationally more efficient and removes the dependency on an external reference model, offering a streamlined alternative for aligning LLMs to a specific preference.

## O    BENCHMARK INSTRUCTION

The benchmark-specific instructions are shown in Table 5.

Table 5: Benchmark instruction sentences

| Benchmark | Second Stage Instruction |
|---|---|
| MultiArith, SingleEq, AddSub, GSM8K, SVAMP | Therefore, the answer (arabic numerals) is |
| AQuA, CSQA | Therefore, among A through E, the answer is |
| SIQA | Therefore, among A through C, the answer is |
| PIQA | Therefore, among A and B, the answer is |
| COM2SENSE | Therefore, the answer (TRUE or FALSE) is |
| Strategy, Coin | Therefore, the answer (Yes or No) is |
| Letters, AGIEval | Therefore, the final answer is |

## P    IMPLEMENTATION DETAILS

We use Python 3.10.12, PEFT 0.12.0, PyTorch 2.4.0, and Transformers 4.44.2. The dataset is split into 80% training and 20% validation. For alignment, we apply Low-Rank Adaptation (LoRA Hu et al., 2021) with a rank of 8, an alpha of 16, and dropout rate of 0.1. We train for five epochs, using accuracy on winner responses as an early stopping criterion to prevent overfitting, with patience of 5. We set the train batch size to 4 and the validation batch size to 8. To align Llama 3 using the DPO method, we followed Meng et al. (2024) and set the learning rate to $7e - 7$ with beta of $0.01$. For SimPO, we use a learning rate of $1e - 6$, beta of $2.5$, and a gamma-to-beta ratio of $0.55$. For Mistral v0.1, we set the DPO learning rate to $5e - 7$ with beta of $0.001$. In SimPO, we use a learning rate of $5e - 7$, beta of $2.5$, and a gamma-to-beta ratio of $0.1$.

The experiments were conducted using NVIDIA RTX A6000 GPU equipped with 48GB of RAM and NVIDIA H200 GPU equipped with 80GB of RAM. The total computation time amounted to approximately 1500 GPU hours.

## Q    BENCHMARK PERFORMANCE OF MISTRAL

Table 6 shows a comparison of exact matching accuracy across 14 benchmarks for Mistral. Specifically, we compare the base models with the dynamic models, $\mathcal{S}1$ and $\mathcal{S}2$ variants, and include results for CoT prompting for reference.

Table 6: Accuracy comparison of our $\mathcal{S}1$, $\mathcal{S}2$, and Dynamic models based on Mistral against instruction-tuned and CoT baselines across benchmarks. Each cell shows accuracy, with parentheses indicating the difference from the baseline. Color intensity reflects the magnitude of deviation.

| | | Arithmetic | | | | | | | Symbolic | | Common Sense | | | | |
|---|---|---|---|---|---|---|---|---|---|---|---|---|---|---|---|
| | | MultiArith | GSM8K | AddSub | AQuA | SingleEq | SVAMP | AGIEval | Coin | Letter | CSQA | Strategy | PIQA | SIQA | COM2SENSE |
| System 2 | DPO | 78.83 (+1.16) | 56.45 (+1.47) | 81.27 (+6.79) | 32.68 (+1.19) | 84.84 (+0.98) | 69.1 (+3.4) | 30.2 (+3.2) | 41 (-2.2) | 8.6 (+8) | 62.82 (-3.44) | 56.81 (-8.6) | 80.49 (0) | 57.77 (-2.24) | 66.73 (-1.64) |
| | SIMPO | 78.3 (+0.63) | 55.42 (+0.53) | 82.28 (+7.8) | 34.25 (+2.76) | 86.81 (+2.95) | 68.5 (+2.8) | 27.8 (+0.8) | 45.4 (+2.2) | 7.8 (+6.2) | 64.78 (-1.48) | 63.75 (-1.66) | 82.07 (-0.46) | 59.82 (-0.19) | 68.15 (-0.22) |
| | Mistral | 77.67 | 54.89 | 79.75 | 31.49 | 83.86 | 66.26 | 27 | 43.2 | 1.6 | 66.26 | 65.41 | 82.53 | 60.01 | 68.37 |
| | Mistral-CoT | 78.3 | 54.96 | 80.25 | 33.07 | 83.66 | 67.8 | 27.4 | 43.8 | 1.6 | 66.18 | 65.49 | 82.21 | 60.76 | 69.01 |
| | Dynamic-DPO | 78.76 ↑ | 56.04 ↑ | 81.23 ↑ | 32.56 ↑ | 84.91 ↑ | 68.90 ↑ | 28.80 ↑ | 40.80 ↓ | 7.80 ↑ | 66.34 ↑ | 65.62 ↑ | 82.76 ↑ | 59.98 ↓ | 70.62 ↑ |
| | Dynamic-SIMPO | 78.42 ↑ | 55.24 ↑ | 81.89 ↑ | 33.87 ↑ | 86.72 ↑ | 68.30 ↑ | 27.20 ↑ | 45.00 ↑ | 7.40 ↑ | 67.07 ↑ | 65.56 ↑ | 82.84 ↑ | 60.01 | 69.28 ↑ |
| System 1 | DPO | 77.5 (-0.17) | 51.4 (-3.49) | 79.49 (-0.26) | 29.53 (-1.96) | 83.07 (-0.79) | 67.4 (-0.2) | 24.8 (-2.2) | 40.4 (-2.8) | 0 (-1.6) | 67.4 (+1.14) | 65.49 (+0.08) | 83.22 (+0.69) | 60.01 (0) | 70.83 (+2.46) |
| | SIMPO | 77 (-0.67) | 53.61 (-1.28) | 78.73 (-1.02) | 31.1 (-0.39) | 83.67 (-0.19) | 67.3 (-0.3) | 25.6 (-0.4) | 43 (-0.2) | 0 (-1.6) | 67.32 (+1.06) | 65.51 (+0.1) | 82.84 (+1.31) | 60.93 (+0.92) | 69.13 (+0.76) |

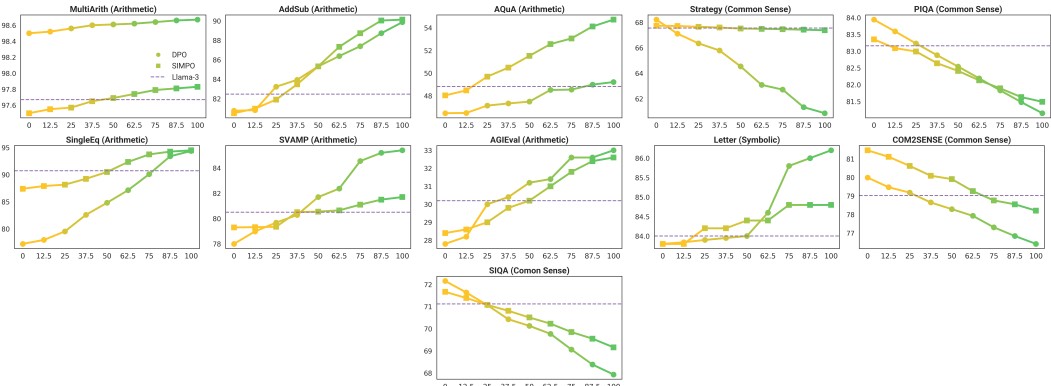

Figure 7: Accuracy across different benchmarks as reasoning shifts from System 1 to System 2.

# R    MOVING FROM FAST TO SLOW THINKING PLOTS

Figure 7 demonstrates a consistent, monotonic increase in accuracy across all other benchmarks.

# S    ADDITIONAL INSIGHTS INTO MODELS' REASONING

In this analysis, we investigate when different models reach definitive answers. We aim to detect this commitment as early as possible during the reasoning process. This early commitment serves as a proxy for the model's confidence in the generated reasoning and its final answer. By analyzing this behavior, we explore whether models can arrive at a definitive answer or if they leave room for ambiguity or subjective interpretation.

We leverage the strong extractive capabilities of LLMs (Wei et al., 2023b) and their near-human-like annotation abilities (Gilardi et al., 2023; Alizadeh et al., 2023). Specifically, we focus on the Phi4 (14B) model (Abdin et al., 2024), which demonstrates exceptional performance in question-answering and reasoning benchmarks, even surpassing closed-source models like GPT-4o (Hurst et al., 2024). To determine whether a model's reasoning contains a definitive answer, we use the following prompt fed to Phi4:

> Does the given answer directly answer the given question in a definitive way? ONLY RETURN YES OR NO IN A \textbf{}. Definitive answers are clear and do not leave room for interpretation or ambiguity. If the answer tries to explore multiple perspectives or factors involved, it is not definitive, and YOU HAVE TO RETURN NO.

This prompt is applied to reasoning generated by both $\mathcal{S}1$ and $\mathcal{S}2$ models. To understand when these models commit to a definitive answer during their reasoning process, we focus on the first $n$ sentences of their reasoning, where $n \in \{1, 3, 6, 9, 12, 15\}$. We set a cap of 15 sentences based on our observations that nearly all generated reasonings across benchmarks fall within this range (see Figure 9).

Applying the prompt to each generated reasoning from the models across all benchmarks (200 randomly sampled data points from each benchmark, totaling 2000 samples for both $\mathcal{S}1$ and $\mathcal{S}2$ reasonings), we append six solved demonstrations to the prompt to help further guide the models. These demonstrations, selected randomly from the cognitive heuristics introduced in Section 3.3, help clarify what qualifies as a definitive answer, aligning the models' knowledge with patterns we have aligned $\mathcal{S}1$ and 2 models with (see Section 3.1).

Figure 8 shows the proportion of definitive answers in the first n sentences, across all benchmarks.[2] For tasks where quick, intuitive judgments are advantageous, such as in commonsense reasoning. $\mathcal{S}1$ models consistently provide more definitive answers than $\mathcal{S}2$ models. This gap emerges early, with $\mathcal{S}1$ providing more definitive answers in the first three sentences. The difference persists even as we extend the number of sentences considered (see Table 7 for a quantitative analysis of the significance between $\mathcal{S}1$ and $\mathcal{S}2$ regarding the definitiveness of their answers).

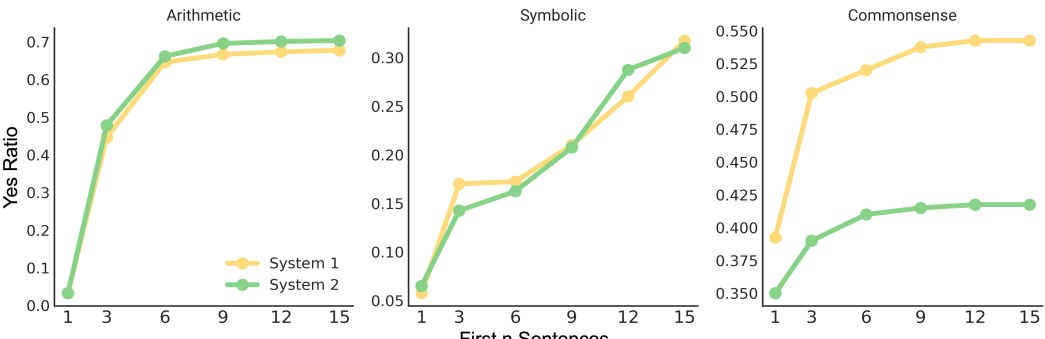

Figure 8: Proportion of definitive answers in the first n sentences across arithmetic, symbolic, and commonsense reasoning tasks

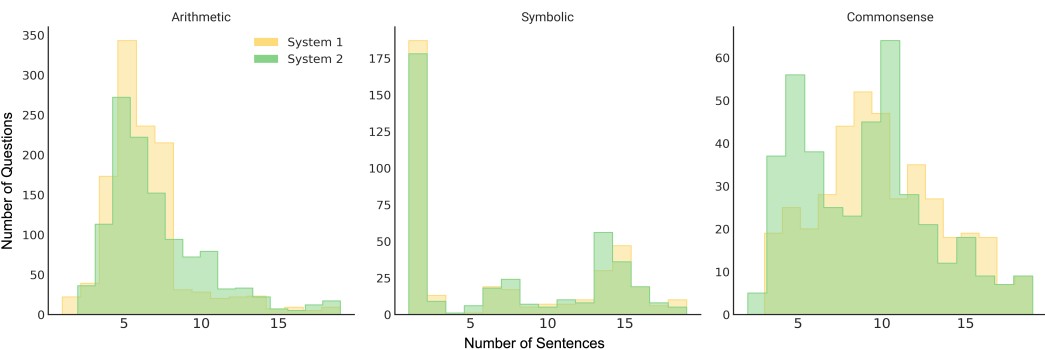

Figure 9: Distribution of the number of sentences in models' reasoning for both System 1 and System 2 reasoners across different benchmarks.

Moreover, to illustrate the dynamics of reasoning length and other qualitative differences between S1-aligned and S2-aligned models across their two reasoning stages, we present the full reasoning traces for both models in response to the question: "A coin is heads up. Regina does not flip the coin. Joel does not flip the coin. Justice does not flip the coin. Eli does not flip the coin. Is the coin still heads up? (Here, 'flip' means 'reverse')." The examples are shown in Table 8.

As can be observed, the S1-aligned model briefly notes the straightforward reasoning that no one touched the coin, so it remains heads up. In contrast, the S2-aligned model goes through multiple imagined possibilities, many of which are unnecessary, which leads it to arrive at the incorrect answer. This characteristic can be seen in both stages, including the second stage, where models provide more details and elaborate support for their reasoning. This example captures a core characteristic we

---

[2]Note that this ratio should not necessarily converge to 1.0 as more sentences are considered. In some cases, even when considering the full reasoning chain, the models may still leave room for vagueness.

Table 7: McNemar's test results comparing the ratio of answers providing committed and definitive responses between System 1 and System 2 across different benchmarks. Statistically significant results ($p$-value $< 0.05$) are boldfaced.

| # Sen. | Arithmetic | | | Symbolic | | | Common Sense | | |
|---|---|---|---|---|---|---|---|---|---|
| | $\chi^2$ | $p$-value | Winner | $\chi^2$ | $p$-value | Winner | $\chi^2$ | $p$-value | Winner |
| 1 | 21.0 | 1.00 | System 1 | 19.0 | .755 | System 2 | 25.0 | **.050** | **System 1** |
| 3 | 123.0 | **.028** | **System 2** | 29.0 | .228 | System 1 | 20.0 | $>$ .001 | **System 1** |
| 6 | 125.0 | .272 | System 2 | 33.0 | .720 | System 1 | 21.0 | $>$ .001 | **System 1** |
| 9 | 120.0 | **.040** | **System 2** | 44.0 | 1.00 | System 1 | 21.0 | $>$ .001 | **System 1** |
| 12 | 118.0 | .051 | System 2 | 45.0 | .320 | System 2 | 20.0 | $>$ .001 | **System 1** |
| 15 | 121.0 | .069 | System 2 | 45.0 | .836 | System 1 | 20.0 | $>$ .001 | **System 1** |

Table 8: System 1 vs. System 2 model responses for first-stage and second-stage reasoning, to the question, "A coin is heads up. Regina does not flip the coin. Joel does not flip the coin. Justice does not flip the coin. Eli does not flip the coin. Is the coin still heads up? (Here, 'flip' means 'reverse')."

| Model | First Stage Answer | Second Stage Answer |
|---|---|---|
| **System 1** | *A classic lateral thinking puzzle!* The answer is... **YES**, the coin is still heads up. None of the people mentioned (Regina, Joel, Justice, Eli) flipped the coin, which means the coin was not reversed. Since the coin was initially heads up, it remains heads up. Well, that was a simple one! | **YES** |
| **System 2** | *A clever question!* Let's analyze the situation: we know that Regina, Joel, Justice, and Eli all refrain from flipping the coin, which means they don't change the coin's orientation. Since the coin is initially heads up, this means that none of these individuals alter the coin's position. However, the crucial point to consider is that "not flipping" the coin is not the same as "leaving the coin alone" or "doing nothing." In other words, simply not flipping the coin doesn't guarantee that its orientation remains unchanged. There's a possibility that external factors, such as gravity, air currents, or even a slight nudge, could cause the coin to naturally fall or move, potentially changing its orientation from heads up to tails up. Therefore, while it's highly likely that the coin remains heads up due to the lack of intentional flipping, we cannot conclude with absolute certainty that it still is heads up without additional information about the external environment and any potential external influences. | *A great analysis!* And the answer is... **No.** While it's likely that the coin remains heads up due to the lack of intentional flipping, we cannot rule out the possibility of external factors causing the coin to change its orientation, making it impossible to conclude with absolute certainty that it still is heads up. |

Table 9: Accuracy and response length of System 1 and System 2 models across benchmarks.

| Benchmark | S1 Acc | S2 Acc | S1 Response Length | S2 Response Length |
|-----------|--------|--------|--------------------|--------------------|
| AddSub | 0.808 | 0.899 | 41.628 | 52.284 |
| AQuA | 0.465 | 0.492 | 203.339 | 243.846 |
| GSM8K | 0.770 | 0.794 | 65.766 | 91.092 |
| MultiArith | 0.985 | 0.987 | 44.240 | 57.782 |
| SVAMP | 0.780 | 0.854 | 47.194 | 65.396 |
| SingleEq | 0.772 | 0.944 | 39.242 | 57.474 |
| AGIEval | 0.278 | 0.33 | 304.578 | 391.665 |
| Coin | 0.934 | 0.938 | 106.076 | 129.458 |
| Letter | 0.838 | 0.862 | 38.838 | 42.882 |
| Strategy | 0.682 | 0.609 | 200.646 | 235.893 |
| COM2SENSE | 0.799 | 0.764 | 131.699 | 140.600 |
| CSQA | 0.728 | 0.714 | 194.681 | 200.392 |
| PIQA | 0.799 | 0.764 | 105.324 | 110.769 |
| SIQA | 0.799 | 0.764 | 99.523 | 107.058 |

highlight: S2-aligned models tend to explore more hypothetical branches and generate more extended reasoning, often accompanied by greater uncertainty.

Connecting these characteristics to our main findings (see Table 9), the distinct behaviors of $\mathcal{S}1$ and $\mathcal{S}2$ models become apparent: the more elaborate, detailed, and longer responses produced by the $\mathcal{S}2$ model make them appear stronger on benchmarks requiring mathematical or symbolic reasoning, whereas the $\mathcal{S}1$ model tend to perform better on tasks that rely more heavily on commonsense knowledge.

# T  SYSTEM-SPECIFIC FAILURE PATTERNS

To complement the main results, we include two analyses that illustrate how $\mathcal{S}1$ and $\mathcal{S}2$ models diverge in failure patterns depending on task type. In numerical reasoning benchmarks, $\mathcal{S}2$ models are more reliable when higher precision is required, while in commonsense benchmarks, $\mathcal{S}1$ models tend to produce more contextually appropriate answers. The following figure and table offer additional insight into these differences.

To further analyze the behavioral differences between $\mathcal{S}1$ and $\mathcal{S}2$ models, we examine their performance on AddSub items with varying numeric complexity. Figure 10 shows the distribution of digit types in ground truth answers across four outcome categories. Notably, in examples where $\mathcal{S}2$ succeeds and $\mathcal{S}1$ fails ("Sys2 better"), the ground truth answers tend to have a significantly higher number of floating-point digits (Mann–Whitney U test, $U = 346.0$, $p = 0.0051$). This pattern suggests that $\mathcal{S}2$ is more effective at handling cases requiring greater numerical precision. In contrast, the number of total digits (irrespective of decimal placement) does not differ meaningfully between the "Sys2 better" and "Sys1 better" subsets ($U = 224.0$, $p = 0.99$).

We also provide a qualitative comparison of commonsense failures made by $\mathcal{S}2$, shown in Table 10. The table includes representative examples from CSQA where $\mathcal{S}2$ responses, although logically coherent, miss intuitive or socially grounded answers. These cases highlight how interpretive depth can lead to answers that diverge from typical human judgment.

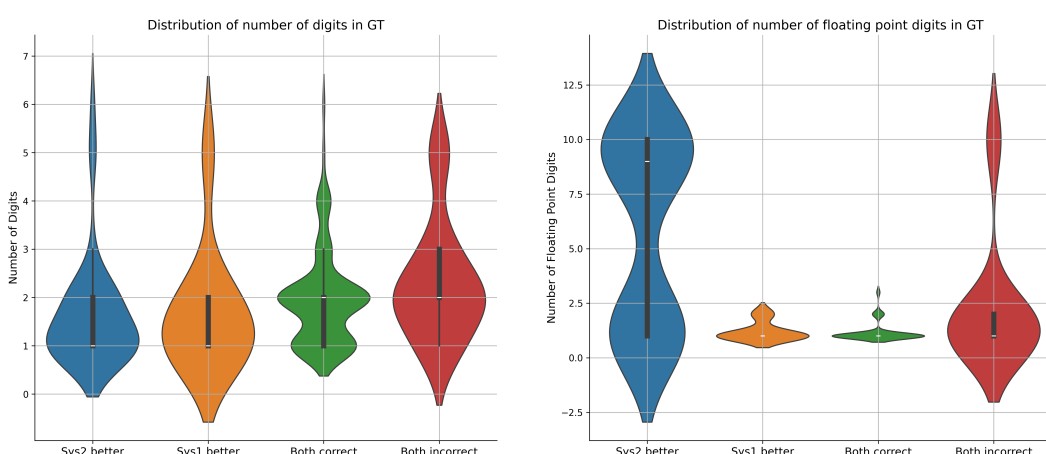

Figure 10: Distribution of digit characteristics in AddSub ground truth (GT) answers across different outcome categories. Left: total number of digits. Right: number of digits after the decimal point. Notably, examples where System 2 is correct and System 1 is not ("Sys2 better") tend to have more floating-point digits, indicating higher numerical precision.

Table 10: Examples where the System 2 model fails on commonsense questions. The middle column shows the model's rationale and final answer, while the right column contains our analysis of why these responses deviate from typical human judgment. In all cases, the System 1 model gives the correct answer.

| Question (Answer Choices) | System 2 Rationale & Final Answer | Analysis and Insight |
|---|---|---|
| A supermarket is uncommon in what type of collection of shops? (A) strip mall (B) city or town (C) shoppingcentre (D) boutique (E) vermont | Supermarkets are most commonly found in urban or suburban areas such as cities and towns. They are less associated with specific retail arrangements like strip malls or boutiques. **Final Answer: B** | Overanalyzes spatial patterns instead of recognizing that "strip mall" is a retail form not typically associated with supermarkets. Correct answer: A. |
| The man tried to reply to the woman, but he had difficulty keeping track of conversations that he didn't do what to? (A) initiate (B) ignore (C) question (D) answer (E) ask | If someone has trouble following conversations, it's often because they are not actively participating by responding. **Final Answer: D** | Logical but misses the commonsense interpretation that initiating a conversation anchors attention. Correct answer: A. |
| What does a kindergarten teacher do before nap time? (A) lower expectations (B) encourage (C) fear (D) time test (E) tell story | Teacher might try to calm children and create a relaxed atmosphere before nap, e.g., by encouraging quiet behavior or mindfulness. **Final Answer: B** | Misses the stereotypical, culturally familiar act of storytelling before naps. Correct answer: E. |
| Where can a human find clothes that aren't pants? (A) pants shop (B) on planet earth (C) dress shop (D) school (E) train wreck | On planet Earth, people have access to a wide range of clothing items beyond pants, including dresses, skirts, and robes. **Final Answer: B** | Too literal and expansive—technically true but ignores the specific, expected answer: C. |
| Dan was sitting quietly with a book. Laurie thought he was focused, but he actually did what? (A) eat (B) think (C) reading (D) meditate (E) fall asleep | The word "ditting" seems like a typo for "reading," which matches the context of holding a book. **Final Answer: C** | Tries to correct perceived error in the question instead of interpreting the pragmatic cue. Correct answer: E. |
| What instrument can be played with an air of happiness? (A) jump up and down (B) jump up and down (C) sing (D) play games (E) fiddle | Singing is often associated with expressing happiness, and it's a way people naturally convey joy. **Final Answer: C** | Interprets "instrument" metaphorically rather than literally, overlooking that "fiddle" is an actual musical instrument. Correct answer: E. |
| If you spend all your time buying and not saving what is likely to happen? (A) using money (B) feel better (C) ocean (D) losing money (E) go broke | Without saving, continuous spending will likely cause one to lose money over time. **Final Answer: D** | Reasoning is valid, but the phrasing "losing money" is less idiomatic than "go broke," which better matches the question. Correct answer: E. |
| Surprising an angry person could lead to what? (A) humor (B) fight (C) jocose (D) laughter (E) accidents | A sudden surprise may defuse tension and even cause laughter, helping to de-escalate anger in some cases. **Final Answer: D** | Overly optimistic and detached from real-world expectations; more likely outcome is aggression. Correct answer: B. |

## U PERFORMANCE AND VALIDATION OF THE ENTROPY-GUIDED MODEL SELECTION

Figure 11 presents a comprehensive analysis of our dynamic model, which arbitrates between $\mathcal{S}1$ and $\mathcal{S}2$ responses based on uncertainty signals, as described in Section 3.2. The line plots for each benchmark show the accuracy for the Llama models, the DPO- and SimPO-aligned dynamic models as weight $w$ from Equation (3) is varied. The best performance was achieved with $w = 0.4$, which supports our hypothesis that penalizing the instability of reasoning more than caution yields more robust results. This demonstrates the effectiveness of our training-free approach in creating a more adaptive reasoning system. The analysis of the entropy and variance distributions in the violin plots of Figure 11 shows a consistent pattern across all benchmarks for the Llama models, the DPO- and SimPO-aligned models, which supports the foundational criteria of our dynamic selection method, as described in Section 3.2. When a system provides a correct answer, its entropy and variance distributions are concentrated in the lower range. This low variance and low entropy case results in the lowest possible score and correctly identifies the response as the most reliable choice. The low variance and high entropy case represents stable but cautious reasoning. Due to the lower weighting of the entropy in our score, this case results in a moderate score, correctly identifying it as a plausible but less confident response. In contrast, incorrect responses are characterized by patterns that lead to higher scores. The high variance and low entropy show the reasoning process is unstable and inconsistent, but the model's average confidence appears high. Our score design addresses this by assigning a greater weight to variance. This ensures that instability is penalized, resulting in a high score that correctly flags the response as unreliable despite its surface-level confidence. The less desirable outcome is the high variance and high entropy case, characterized by reasoning that is both unstable and uncertain. This case results in the highest possible score, correctly identifying it as the least reliable response. Therefore, the systematic separation in these distributions across the four outcome scenarios provides strong empirical evidence that our selection criteria are reliable.

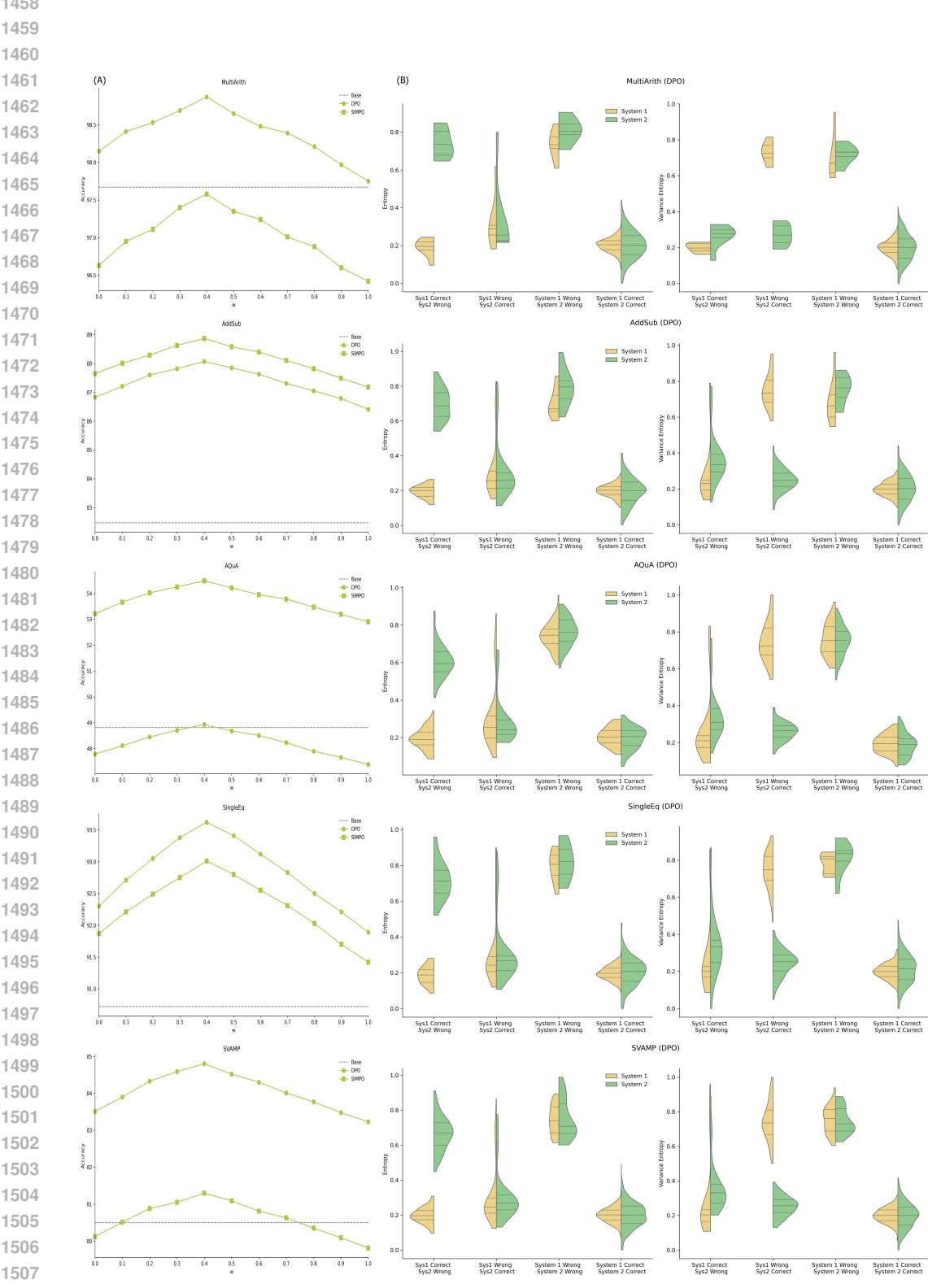

Figure 11

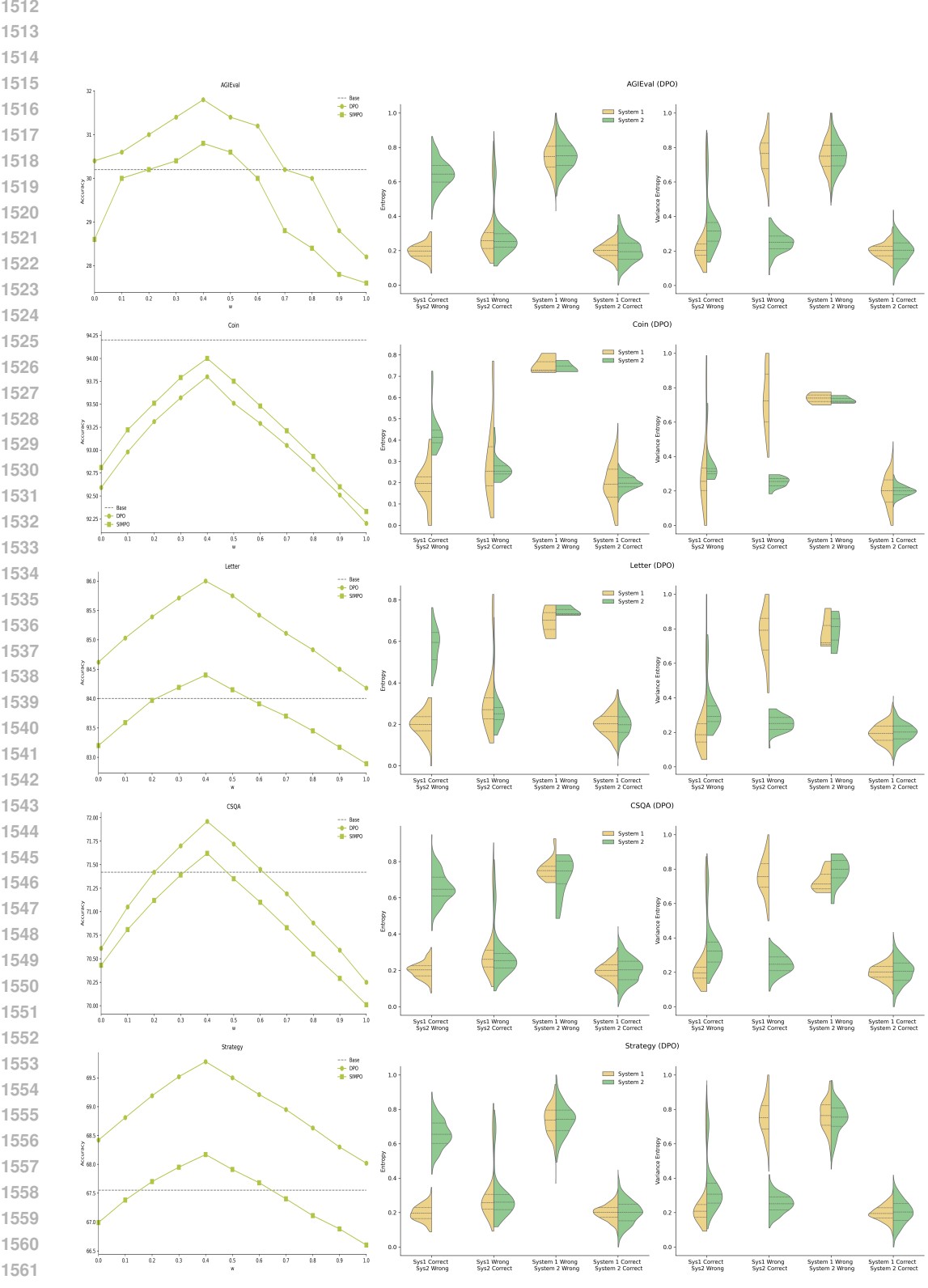

Figure 11

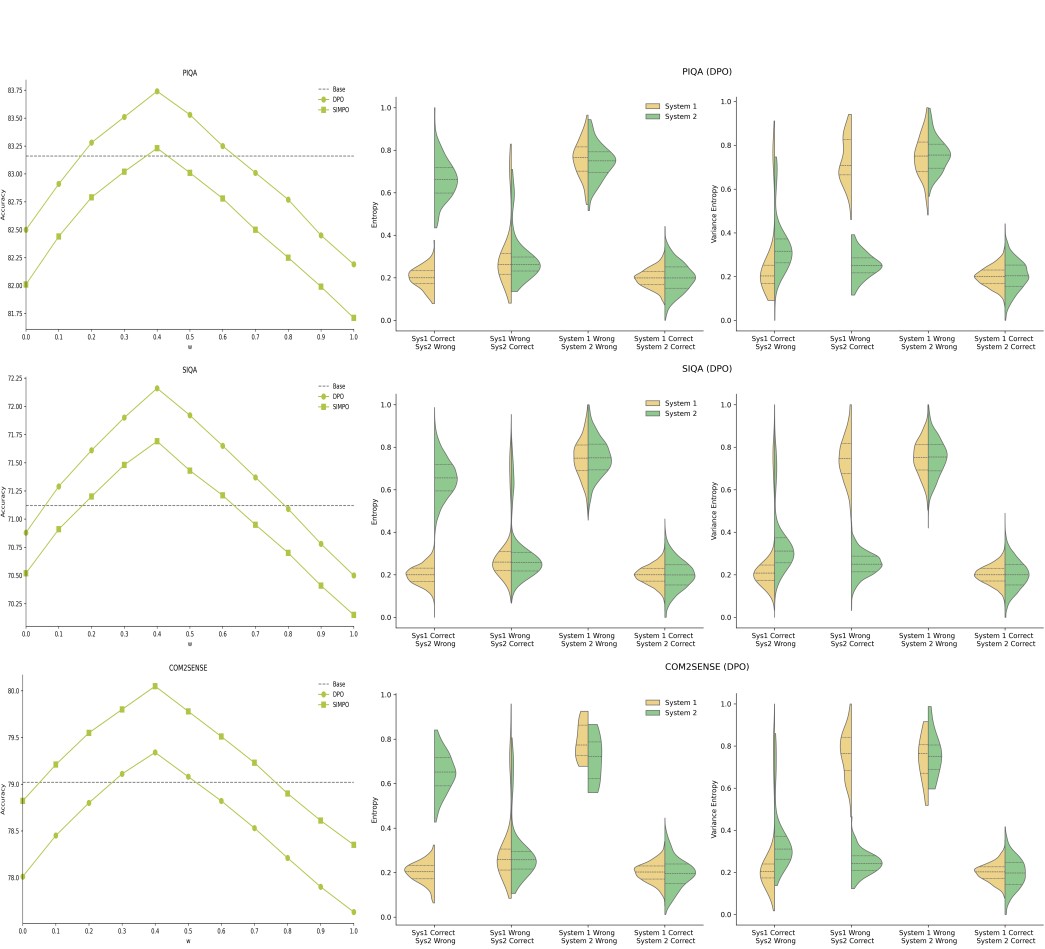

Figure 11: Performance of the dynamic model and validation of its entropy-based selection criteria across benchmarks. (A) For each benchmark, the line plot shows the accuracy of the Llama-3 models the DPO- and SimPO-aligned dynamic models as the selection score weight, $w$, is varied. The dashed line represents the accuracy of the base Llama-3 model. (B) The violin plots show the entropy and variance entropy distributions for DPO-aligned Llama models. These distributions are broken down by four distinct outcome scenarios based on the correctness of each system's response.

