# OpenReview forum: "Reasoning on a Spectrum: Aligning LLMs to System 1 and System 2 Thinking"
_ICLR.cc/2026/Conference — Submitted to ICLR 2026_

### Official Review · Reviewer_rtgL · 2025-10-23

**Soundness:** 4
**Presentation:** 4
**Contribution:** 3
**Rating:** 6
**Confidence:** 5

**Summary:**

This paper, grounded in human cognitive theory, examines how System‑1 and System‑2 data influence large language models and analyzes the characteristics of S1/S2‑aligned models. It further introduces a training‑free, entropy‑based dynamic model selection method that improves answer accuracy. The motivation of the article is clear and the experimental part is also sufficient.

**Strengths:**

1. This article is well written, the concepts and formulas involved in the article are explained very clearly and the motivation is clearly expressed, which make it easy to follow.
2. The experimental part of this article is very thorough, with a large number of experiments and verification on the data set, making the article more convincing.

**Weaknesses:**

I'm curious about **line 242**. The article mentions a two-stage process. In the first stage, the answer lengths of S1 and S2 are similar, but after the second stage, S2's answer length is higher than S1's. Although the experiment is clearly described in the article, and I've looked at the prompt in Table 5 ("Therefore, xxx, the final answer is"), I still find it a bit strange why simply adding such a prompt increases the difference in answer length between S1 and S2. Could the authors provide a concrete example to illustrate this? Could you also provide a table showing the accuracy and number of tokens for the S1 and S2 models on GSM8K, Coin, and CSQA for a more intuitive understanding?

**Questions:**

Currently, most people believe that the difference between System 1 and System 2 models is length, which is largely due to the success of DeepSeek-R1. However, this article gives us a different perspective. The length of the answers in the first stage of S1 and S2 is similar. If the answer involves a lot of reflection (such as **aha moments** observed in DeepSeek), the answer length will inevitably be long. Have you observed similar phenomena in your experiments?

---

> ### Author Response · Authors · 2025-11-16
>
> We thank the Reviewer for finding our paper well-motivated and well-executed in its methodology and experiments. We have also conducted additional analyses to address the question you raised about the difference in response length between System 1- and System 2-aligned models.
>
> To elaborate on our findings, based on our observations, the S2-aligned models overall produced longer responses compared to the S1-aligned models. Using our two-stage evaluation framework, we observed a marginally significant difference in the first stage, where models provide their reasoning and justification for their answers, t(9336) = 1.71, p = 0.087, Cohen’s d = 0.02 (using LLaMA + DPO for instance), and a significant difference in the second stage, where models reiterate on their first-stage reasoning and produce a final answer: t(9336) = 56.109, p < 0.001, Cohen’s d = 0.58. This distinctive behavior of S2-aligned models appears in the way they reiterate and elaborate on their initial reasoning when finalizing their answers, exhibiting greater verbosity and uncertainty, an attribute we identify as a core characteristic of System-2 alignment in Section 5.3.
>
> To better illustrate this, we present below the reasoning produced by S1-aligned and S2-aligned models in Stage 1 and Stage 2 for the following question:
>
> A coin is heads up. Regina does not flip the coin. Joel does not flip the coin. Justice does not flip the coin. Eli does not flip the coin. Is the coin still heads up? Note that "flip" here means "reverse".
>
>
> | **System** | **First Stage Answer** | **Second Stage Answer** |
> |-------------|------------------------|--------------------------|
> | **System 1** | *A classic lateral thinking puzzle!* The answer is... **YES**, the coin is still heads up. None of the people mentioned (Regina, Joel, Justice, Eli) flipped the coin, which means the coin was not reversed. Since the coin was initially heads up, it remains heads up. Well, that was a simple one! | **YES** |
> | **System 2** | *A clever question!* Let's analyze the situation: we know that Regina, Joel, Justice, and Eli all refrain from flipping the coin, which means they don't change the coin's orientation. Since the coin is initially heads up, this means that none of these individuals alter the coin's position. However, the crucial point to consider is that "not flipping" the coin is not the same as "leaving the coin alone" or "doing nothing." In other words, simply not flipping the coin doesn't guarantee that its orientation remains unchanged. There's a possibility that external factors, such as gravity, air currents, or even a slight nudge, could cause the coin to naturally fall or move, potentially changing its orientation from heads up to tails up. Therefore, while it's highly likely that the coin remains heads up due to the lack of intentional flipping, we cannot conclude with absolute certainty that it still is heads up without additional information about the external environment and any potential external influences. | *A great analysis!* And the answer is... **No.** While it's likely that the coin remains heads up due to the lack of intentional flipping, we cannot rule out the possibility of external factors causing the coin to change its orientation, making it impossible to conclude with absolute certainty that it still is heads up. |
>
> As can be observed, the S1-aligned model briefly notes the straightforward reasoning that no one touched the coin, so it remains heads up. In contrast, the S2-aligned model goes through multiple imagined possibilities, many of which are unnecessary, which leads it to arrive at the incorrect answer. This characteristic can be seen in both stages, including the second stage, where models provide more details and elaborate support for their reasoning. This example captures a core characteristic we highlight: S2-aligned models tend to explore more hypothetical branches and generate more extended reasoning, often accompanied by greater uncertainty.

---

> ### Author Response · Authors · 2025-11-16
>
> Further, regarding your suggestion to provide a more intuitive visualization of accuracy and response length for both systems, please find the requested table below (again using LLaMA + DPO):
>
> | Benchmark | S1 Acc | S2 Acc | S1 Response Length | S2 Response Length |
> |-----------|--------|--------|---------------------|---------------------|
> | AddSub    | 0.808  | 0.899  | 41.628              | 52.284              |
> | AQuA      | 0.465  | 0.492  | 203.339             | 243.846             |
> | GSM8K       | 0.770  | 0.794  | 65.766              | 91.092              |
> | MultiArith     | 0.985  | 0.987  | 44.240              | 57.782              |
> | SVAMP     | 0.780  | 0.854  | 47.194              | 65.396              |
> | SingleEq    | 0.772  | 0.944  | 39.242              | 57.474              |
> | Coin      | 0.934  | 0.938  | 106.076             | 129.458             |
> | Letter    | 0.838  | 0.862  | 38.838              | 42.882              |
> | Strategy  | 0.682  | 0.609  | 200.646             | 235.893             |
> | COM2SENSE    | 0.799  | 0.764  | 131.699             | 140.600             |
> | CSQA    | 0.728  | 0.714  | 194.681             | 200.392             |
> | PIQA    | 0.799  | 0.764  | 105.324              | 110.769               |
> | SIQA    | 0.799  | 0.764  | 99.523             | 107.058             |
>
> The above table reflects the findings of Sections 5.1 and 5.2, reiterating that S1-aligned models tend to perform better on tasks requiring commonsense knowledge, while S2-aligned models outperform S1-aligned models on tasks requiring mathematical or more symbolic reasoning. We revised the article to include both tables (Table 7 & 8) in the appendix to improve clarity and provide better examples for readers.
>
> Finally, regarding your question about differences in the reasoning characteristics between the first and second stages: as noted earlier, we observed longer responses from S2-aligned models, along with qualitatively different reasoning strategies marked by higher uncertainty and a tendency to consider an excessive number of hypothetical possibilities. This aligns with your remark that the difference cannot be reduced solely to answer length.
>
> In relation to the “Aha moment” phenomenon emerging in large reasoning models such as DeepSeek-R1, our results support an interesting parallel: S2-aligned models also exhibit longer, more elaborative, and detailed reasoning chains. However, the origins differ. Our S2-aligned models are explicitly trained to emulate cognitive-science-inspired System 2 thinking which is the target of our study alongside all the relevant characteristics like higher level of uncertainty, and not necessarily R1-type reflections. However, the extended reasoning chains in large RL-trained reasoning models arise from large-scale reinforcement learning with verifiable reward signals, leading the models to iterate over correct and incorrect chains to reach a solution, and thereby increasing reflections compared to non-reasoning fine-tuned models [1].
>
> [1] Yang, S., Wu, J., Chen, X., Xiao, Y., Yang, X., Wong, D. F., & Wang, D. (2025). Understanding aha moments: from external observations to internal mechanisms. arXiv preprint arXiv:2504.02956.

---

> > ### Comment · Reviewer_rtgL · 2025-11-21
> > **Reply**
> >
> > Thank you for your reply, I have no further questions. Overall, I think this work is very informative and provides a lot of analysis on system1/2; it is a solid piece of work.

---

> > > ### Author Response · Authors · 2025-12-01
> > > **Confirmation of Score Update After Rebuttal**
> > >
> > > Thanks for your reply and for increasing your overall score. As you know, due to the OpenReview issue and the decision made by the ICLR venue accordingly, the scores will revert to what they were before the rebuttal, and new ACs are being assigned; there will be no visibility of the changed score. Since your response did not explicitly mention the change, would it be possible for you to confirm here that you updated your score after our discussion so that the new AC can be aware of the result of our rebuttal?

---

### Official Review · Reviewer_2qiX · 2025-10-25

**Soundness:** 3
**Presentation:** 3
**Contribution:** 3
**Rating:** 6
**Confidence:** 3

**Summary:**

This paper conducts a series of experiments and analyses on System 1 and System 2 reasoning paradigms. The authors argue that each has its own strengths: System 1 is efficient for heuristic problem-solving, while System 2, by introducing controlled uncertainty to support reasoning, is better suited for tasks requiring rigorous logical inference. To address the question of when to use each reasoning mode, the authors train two types of models and introduce a reliability metric to determine which model is more appropriate for a given task.

**Strengths:**

- The paper is well motivated, and the experimental design is thorough.
- The results align with expectations, and the authors conduct extensive statistical analyses to ensure their reliability.

**Weaknesses:**

- There are some contradictions in the paper. In Section 3.1, the authors state that LLMs can be guided toward either S1 or S2 reasoning through prompt engineering. However, in the experiments, they instead use DPO to explicitly train models to prefer one reasoning paradigm over the other.
- The reliability metric introduced in Section 3.2 lacks sufficient justification. Specifically, entropy and variance are not intuitively compatible quantities, and weighting and summing them together may undermine their original semantic meaning.
- Regarding the question of when to adopt the more appropriate reasoning paradigm, the proposed approach is not very efficient. It relies on training both S1 and S2 models and then selecting between them using the metric, which increases computational cost.

**Questions:**

In addition to the weaknesses noted above, there are several further questions:

- For the two types of models used in the experiments, the instruction-tuned version and the CoT version, the experimental results are basically consistent. I would like to ask if you have tested the results after using the few-shot version?
- As in 3.3, where the overly long S2 response was shortened, can there be a case to show S1 and S2 responses to a question when their lengths are similar?
- I actually still want to ask why you need to shorten the answer in S2 in 3.3. Can this shortened answer ensure a reasoning effect similar to before? For example, if we input the two types of reasoning steps into LLMs and then force the output answer. Are their answers consistent?

---

> ### Author Response · Authors · 2025-11-16
>
> We thank the Reviewer for recognizing our paper’s motivation, thorough experimental design, and robust statistical validation. We address each concern below.
>
> Weaknesses Response:
>
> 1. We thank the Reviewer for this observation. To be perfectly clear, our method relies entirely on alignment training (DPO/SimPO), not prompt engineering. Our reference to prompt engineering in Section 3.1 was intended as motivation, not a description of our method. The fact that prior work can elicit S1/S2 behaviors with prompts simply demonstrates that these reasoning modes are already latent in pretrained LLMs. Our work builds on this insight. We use alignment training precisely because it goes a step further: it doesn't just elicit a style, it explicitly sharpens, separates, and stabilizes these capabilities, making them consistent and controllable for our experiments. We realize this was not as clear as it could be. To remove this ambiguity, we have revised the text and added the following clarification to line 154: “Given that both modes are already latent in pretrained LLMs, aligning the model to these two styles simply sharpens and separates capabilities that naturally coexist.”
>
> 2. We thank the Reviewer for the helpful comment. We recognize that the original wording may have suggested an intuitive mismatch between entropy and variance. We have revised the text to clarify that our reliability metric does not combine a single entropy and variance term directly. Instead, as defined in Equation (2), it uses two complementary token-level statistics: the average entropy, representing overall uncertainty, and the variance of entropy, capturing instability in the reasoning process. Together, these quantify how confident and consistent a model’s reasoning is. Prior work has shown that combining these signals provides a reliable indicator of reasoning stability and quality [1].
>
> 3. We agree with the Reviewer’s point; we acknowledge this in Appendix A (Limitation), lines 865-867, that our training-free dynamic approach doubles the inference costs, and that future work could investigate more efficient approaches, such as distilling both reasoning modes into a single adaptive model. To further emphasize this limitation, we have now added an explicit statement in the Conclusion (lines 476-480) that highlights the increased inference cost and single model distillation as an important direction for future work.

---

> ### Author Response · Authors · 2025-11-16
>
> Question Responses:
>
> 1. We thank the Reviewer for this suggestion. We did as the Reviewer suggested. We conducted an additional few-shot CoT experiment across all benchmarks. The results are shown in the Table below, comparing the few-shot baseline to the previous baselines. Overall, few-shot prompting tends to improve performance. Across 13 benchmarks, few-shot prompting improved performance in 12, although the gains were relatively modest. Nevertheless, including few-shot CoT as an additional baseline, S2-aligned models still excel in arithmetic and symbolic tasks compared to baselines and S1-aligned models, while S1-aligned models are superior in commonsense reasoning tasks.
>
> | benchmark | base | zero-shot CoT | few-shot CoT | System 1 | System 2 |
> |--- |--- |--- |--- |--- |--- |
> | MultiArith | 97.67 | 97.83 | 98.63 | 98.5 | 98.67 |
> | GSM8K | 78.49 | 78.54 | 79.15 | 77.01 | 79.37 |
> | addsub | 82.47 | 82.03 | 86 | 80.76  | 89.87 |
> | AQuA | 48.82 | 49.21 | 49.86 | 46.46 | 49.21 |
> | SingleEq | 90.72 | 88.19 | 92.21 | 77.24 | 94.37 |
> | SVAMP | 80.5 | 80.9 | 81.5 | 78 | 85.4 |
> | Coin | 94.2 | 94.8 | 94.8 | 93.4 | 93.8 |
> | Letter | 84 | 84.2 | 85 | 83.8 | 86.2 |
> | CSQA | 71.42 | 71.58 | 72.19 | 72.81 | 71.42 |
> | Strategy | 67.55 | 67.38 | 66.98 | 68.21 | 60.87 |
> | PIQA | 83.16 | 83.34 | 83.67 | 83.94 | 81.15 |
> | SIQA | 71.12 | 70.97 | 71.42 | 72.16 | 67.93 |
> | COM2SENSE | 79.02 | 79.86 | 79.93 | 79.99  | 76.42 |
>
>
> 2. Yes. Appendix H, Table 4 shows a sample from our final, curated dataset. These examples all demonstrate S1 and S2 responses that have similar lengths but essentially different reasoning styles, which is one of the key contributions of our paper.
>
> 3. To address the Reviewer’s concern, we conducted an additional ablation study by training our models on the un-normalized data and we have added the results in section 5.2 and appendix M. The analysis from this study confirms that S2 responses are significantly longer than S1 responses, even when length is not normalized (DPO t(8836) = 71.831, p < .001; SIMPO t(8586) = 15.227, p < .001). Moreover, our results in Section 5.2 also demonstrate that S2-aligned models generated significantly longer responses despite being trained on a length normalized dataset (DPO t(8836) = 57.14, p < .001; SimPO t(8586) = 9.833, p < .001). These suggest that the “overthinking” behavior is inherent to the S2 reasoning style; in both of the settings, the S2 models generate more tokens compared to their S1 counterparts. As noted in Lines 210-212, prior work has shown that alignment methods can rely on superficial cues such as output length, which may bias models toward longer responses even without genuine reasoning improvement [2]. Therefore, our primary motivation for length normalization was to remove this superficial cue and ensure our models were learning the reasoning process itself, not just mimicking a long or short output style. Moreover, as noted in Lines 203–206 and shown in Appendix H, our dataset is process-oriented, explicitly focusing on the articulation of S1- and S2-type reasoning styles. This design was implemented to ensure that the reasoning effect was preserved, and our experts verified that the length adjustment only pruned redundant elaborations of the process, not the S2 reasoning structure.
>
> [1] Zhang, Jinghan, et al. "Entropy-based exploration conduction for multi-step reasoning." arXiv preprint arXiv:2503.15848 (2025).
> [2] Singhal, Prasann, et al. "A long way to go: Investigating length correlations in rlhf." arXiv preprint arXiv:2310.03716 (2023).

---

> > ### Comment · Reviewer_2qiX · 2025-11-25
> >
> > Thanks for the detailed reply. I have no further questions at this time and will maintain the positive score.

---

### Official Review · Reviewer_B6EU · 2025-10-26

**Soundness:** 2
**Presentation:** 1
**Contribution:** 2
**Rating:** 2
**Confidence:** 4

**Summary:**

The authors explicitly align LLMs to these reasoning styles by curating a dataset containing valid System 1 and System 2 answers, and evaluate model performance across reasoning benchmarks. Their results reveal an accuracy-efficiency trade-off: System 2-aligned models demonstrate superior performance in arithmetic and symbolic reasoning, while System 1-aligned models achieve better results in commonsense reasoning tasks. To analyze the reasoning spectrum, they interpolate between the two extremes by varying the proportion of alignment data, observing a corresponding monotonic change in accuracy. A mechanistic analysis of model responses indicates that System 1 models tend to produce more definitive outputs, whereas System 2 models exhibit greater uncertainty. Building on these findings, the authors further combine System 1- and System 2-aligned models based on the entropy of their generations without requiring additional training, obtaining a dynamic model that outperforms individual models across nearly all benchmarks.

**Strengths:**

The authors' methodology section is clearly articulated, well-supported by diagrams, and validated through comprehensive experiments.

**Weaknesses:**

1. While one key contribution is "a dynamic model that outperforms across nearly all benchmarks," the main text provides no quantitative data to indicate the magnitude of improvement over previous methods. The authors should include specific performance gain figures whenever claiming superiority over existing approaches.

2. The methodology is primarily demonstrated through Figure 5 showing GSM8K performance, while most experimental results are presented in the appendix in graphical form, making it difficult for readers to quantitatively appreciate the advantages. It is recommended to add an "Ours" row in Table 1 to clearly showcase the method's superiority.

3. According to Figures 5 and 11, the parameter w appears to be separately tuned for each dataset, essentially trying all hyperparameters w for every dataset. This constitutes an unfair evaluation in benchmark testing. Since the method is described as a "dynamic model," there should be a reasonable approach for dynamic adjustment that uses consistent parameters across all evaluations for fair comparison.

4. The models employed by the authors appear somewhat outdated. It is recommended to incorporate more recent model to demonstrate state-of-the-art performance.

5. It is recommended to improve the paper's writing, as the article contains substantial redundant and repetitive information while lacking crucial illustrative examples and quantitative data support. For example, the authors devote excessive length (nearly a full page) to the conclusion section, with substantial portions being repetitive and redundant. Meanwhile, significant experimental content is relegated to the appendix, which undermines the substantive contributions in the main text.

**Questions:**

Same as above

---

> ### Author Response · Authors · 2025-11-16
>
> We thank the Reviewer for constructive feedback.
>
> 1,2. As the Reviewer suggested, we have added the results of the dynamic model across all benchmarks to Table 1 for each model family and alignment method used in our experiments. These additions provide the quantitative evidence requested regarding the magnitude of improvement.
>
> The updated table shows that our dynamic model outperforms the base model on 12 out of 13 benchmarks, and consistently narrows the gap with the best-performing aligned model (either System-1 or System-2) on every benchmark. This demonstrates that the dynamic model provides a strong balance between the two reasoning styles, effectively leveraging the strengths of both.
>
> 3. The dynamic model is not tuned separately for each dataset. The dynamic model that we have presented its results in Table 1 is the same across all benchmarks (using w = 0.4). We initially plot the full performance curve w.r.t w parameter for every benchmark to demonstrate the sensitivity of the method and to verify that the same trade-off between uncertainty and instability governs behavior across tasks, and the optimal region consistently centers around w = 0.4 across all benchmarks. The takeaway of these curves across very different datasets shows that the choice of w depends on the intrinsic behavior of the S1 and S2 models rather than on dataset-specific patterns. We have now included all these figures in the Appendix.
>
> 4. While it is true that more recent models could achieve higher absolute performance, the primary contribution of our paper is not to demonstrate state-of-the-art results on any particular benchmark, but to study the differences between S1 and S2 reasoning strategies: their qualitative characteristics, behavioral signatures, and quantitative performance gaps across diverse benchmarks. Motivated by the Reviewer’s feedback, we also evaluated our method on two additional model variants (3B and 70B). The observed S1/S2 behaviors are consistent across architectures (Llama and Mistral), model scales (3B, 8B, 70B), and algorithms (DPO, SIMPO), indicating that these phenomena are not model-, scale-, or algorithm-specific. These patterns can be reliably observed in any model capable of generating reasoning traces, such as those exhibiting chain-of-thought or related emergent capabilities. Thus, we emphasize our contribution lies in analyzing the characteristics of System 1 and System 2 reasoning, rather than pursuing the highest leaderboard scores.
>
> 5. We appreciate the Reviewer’s recommendation. As noted in our responses to the first two weaknesses, we have now added quantitative results for the dynamic model into Table 1, ensuring that the main text clearly provides the magnitude of performance improvements for dynamic models. In addition, we have run two other model variants (3B-70B) as requested by other Reviewers and will include their results in the revised submission to provide even stronger quantitative support. Since the page limit for the rebuttal phase is 10 pages, we would be very grateful for the Reviewer's specific guidance, given these additions: are there other vital tables or figures from the appendix that you believe are essential to move into the main paper? Then, if we exceed the page limit, we are prepared to shorten the conclusion accordingly to ensure that the most important experimental results appear in the main text.

---

> > ### Author Response · Authors · 2025-11-21
> > **Improve writing and present more quantitative results.**
> >
> > We have updated our paper to include new results for the S1, S2, and dynamic models across both alignment algorithms, evaluated on smaller and larger model scales. Table 1 in the current version now presents Llama 3 results at different scales (3B, 8B, 70B) on reasoning benchmarks. For clarity and organization, the results for the Mistral model have been moved to Table 6 in Appendix Q.

---

### Official Review · Reviewer_AWgf · 2025-11-01

**Soundness:** 3
**Presentation:** 3
**Contribution:** 2
**Rating:** 4
**Confidence:** 4

**Summary:**

This paper explores aligning LLMs with System 1 and System 2 reasoning. It creates a dataset of 2k questions, each paired with responses of both reasoning styles, and normalize their lengths. Using DPO and SimPO, it finds that System 2 aligned models excel at arithmetic and symbolic reasoning, while System 1 aligned models perform better on commonsense tasks. The paper further introduces an entropy-based dynamic model that adaptively selects between the two reasoning modes without additional training.

**Strengths:**

1. The experiments clearly show distinct trends between System 1 and System 2 aligned models, supporting their claims on the difference across different tasks.

2. The performance transition when interpolating between the two reasoning styles is particularly interesting, showing how alignment can gradually shift reasoning behavior from S1 to S2.

3. The writing is clear and well-organized, providing insightful observations on the differing alignment behaviors of the two reasoning modes.

**Weaknesses:**

1. The refinement of matching response lengths between System 1 and System 2 outputs could unintentionally change their natural characteristics. Expanding concise answers into longer forms may introduce redundancy and reduce the intuitive nature of System 1 reasoning. It would be helpful to see whether training without this adjustment yields consistent findings.
2. The proposed entropy-based approach requires running both System 1 and System 2 models and then selecting the final answer, which is an ensemble rather than an adaptive reasoning switch. This limits the claimed cognitive analogy and introduces more inference cost.
3. The math datasets used involve relatively few reasoning steps. Their simplicity may mask the deeper distinctions between System 1 and System 2 reasoning modes.

**Questions:**

Is there any evaluation of whether the aligned reasoning behaviors transfer out of distribution? For instance, does System 2 alignment on math tasks generalize to other domains, and how does it affect performance?

---

> ### Author Response · Authors · 2025-11-16
>
> We thank the Reviewer for finding our experiments clear and interesting. Below, we address the concerns raised.
>
> Weaknesses Response:
>
> 1. We thank the Reviewer for suggesting training without length normalization. We conducted further analysis addressing this point and we have added the results in section 5.2 and appendix M. The analysis confirms that S2 responses are significantly longer than S1 responses, even when the length of the data is not normalized (DPO t(8836) = 71.831, p < .001; SIMPO t(8586) = 15.227, p < .001). Moreover, our results in Section 5.2 also demonstrate that S2-aligned models generated significantly longer responses despite being trained on a length-normalized dataset (DPO t(8836) = 57.14, p < .001; SimPO t(8586) = 9.833, p < .001). These results suggest that the “overthinking” behavior is inherent to the S2 reasoning style; in both settings, the S2 models generate more tokens compared to their S1 counterparts. As noted in Lines 210-212, prior work has shown that alignment methods can rely on superficial cues such as output length, which may bias models toward longer responses even without genuine reasoning improvement [1]. Therefore, our primary motivation for length normalization was to remove this superficial cue and ensure our models were learning the reasoning process itself, not just mimicking a long or short output style. Moreover, as noted in Lines 203–206 and shown in Appendix H, our dataset is process-oriented, explicitly focusing on the articulation of S1- and S2-type reasoning styles. This design was implemented to ensure that the reasoning effect was preserved, and our experts verified that the length adjustment only pruned redundant elaborations of the process, not the S2 reasoning structure.
>
> 2. We agree with the Reviewer’s observation, and we note in Appendix A (Limitation, lines 865-867) that our approach is indeed an ensemble one, and that this training-free dynamic approach doubles the inference costs, and mention that future work can distill both reasoning modes into a single adaptive model. To further clarify this nuance and that this ensemble dynamic approach is adaptive to the models’ confidence and stability, based on which we choose the final reasoning, we have now added a clear statement in the Conclusion (lines 476-480) that highlights the increased inference cost and single model distillation as an important direction for future work.
>
> 3. We appreciate this point. Our goal is not to test the model's maximum mathematical capacity, but to isolate the effect of its reasoning style. Using highly complex benchmarks (such as AIME or MATH) with 8B models would introduce a confound: the models would fail due to their fundamental capability limits, making it impossible to distinguish failures of style (S1 vs. S2) from failures of capacity. We chose benchmarks like GSM8K and AddSub precisely because the base model has the corresponding capabilities, allowing us to observe the performance trade-offs from our alignment. The fact that we already observe a consistent S2 advantage on these simpler tasks does not obscure the distinction; rather, it provides a clean setting in which the effect emerges without being overshadowed by capacity limitations. We further agree with the reviewer's intuition: this S2 advantage would almost certainly be amplified on more complex, multi-step problems. Those tasks explicitly demand the deliberative processing that S2 alignment encourages, further reinforcing our central claim.
>
>
> Questions Response:
>
> We note that all the reported results are based on out-of-distribution transfer. Specifically, we align the models with System 1 or System 2 thinking using a dataset constructed to capture the different reasoning strategies associated with these modes of thinking, as discussed in Section 3.3. This alignment process produces exactly two models: a System-1-aligned model and a System-2-aligned model. Importantly, these models are not trained on any of the downstream benchmarks. Instead, they are only evaluated directly, zero-shot, on all 13 benchmarks presented in the paper. Therefore, all of our reported results already reflect out-of-distribution transfer: the models are aligned only on our custom reasoning-style dataset, and their performance on math, commonsense, symbolic reasoning, and other domains emerges without any task-specific fine-tuning.
>
> [1] Singhal, Prasann, et al. "A long way to go: Investigating length correlations in rlhf." arXiv preprint arXiv:2310.03716 (2023).

---

> > ### Author Response · Authors · 2025-11-21
> > **Reasoning with more steps!**
> >
> > To further address the Reviewer’s concern regarding the simplicity of our math benchmarks (weakness 3) and the need to have reasoning benchmarks with more steps, we expanded our evaluation to include AGIEval [2]. This benchmark requires more complex reasoning, averaging 348.1 tokens per response, compared to 168.2 tokens for the other math benchmarks used in our experiments. The results for Llama-3-8B with DPO alignment are shown below:
> >
> > | Model        | Accuracy |
> > |--------------|----------|
> > | S2-aligned (DPO)   | 33.0     |
> > | Llama 8B         | 30.2     |
> > | S1-aligned (DPO)  | 27.8     |
> >
> > The S2-aligned model outperforms both the base and S1 models. Crucially, the distinction in reasoning depth is more pronounced on this complex task: the gap in average output length between S1 (304.6 tokens) and S2 (391.7 tokens) is noticeably wider than in simpler math tasks.
> >
> > [2] Zhong, Wanjun, et al. "Agieval: A human-centric benchmark for evaluating foundation models." Findings of the Association for Computational Linguistics: NAACL 2024. 2024.

---

> > > ### Author Response · Authors · 2025-11-25
> > > **AGIEval has been added to the paper**
> > >
> > > We have updated our paper to include new results for the AGIEval dataset in Table 1 of the current version. Since adding this dataset affects all reported numbers, several values in the paper are now different from those in our original response. However, these differences are minor, and all conclusions remain unchanged. For your convenience, we report the updated results in the table below:
> > > | Model Size | Variation | Accuracy |
> > > |--------------|----------|----------|
> > > | Llama 70B | S2 (DPO)   | 47.2     |
> > > | Llama 70B |   Base         | 45.8     |
> > > | Llama 70B | S1 (DPO)   | 41.4     |
> > > | Llama 8B   | S2 (DPO)   | 33.0     |
> > > | Llama 8B   |   Base        | 30.2     |
> > > | Llama 8B   | S1 (DPO)   | 27.8     |
> > > | Llama 3B   | S2 (DPO)   | 27.2     |
> > > | Llama 3B   |   Base    	    | 24.6     |
> > > | Llama 3B   | S1 (DPO)   | 22.0     |

---

### Official Review · Reviewer_6SYb · 2025-11-01

**Soundness:** 3
**Presentation:** 4
**Contribution:** 3
**Rating:** 6
**Confidence:** 2

**Summary:**

This paper investigates whether LLMs should always use deliberate reasoning (System 2) or adapt their approach like humans do. The authors trained models to use either fast, intuitive reasoning (System 1) or slow, deliberative reasoning (System 2) using a dataset based on cognitive heuristics. Testing across 13 benchmarks revealed a trade-off: S2 models excelled at arithmetic and symbolic tasks but generated longer responses, while S1 models were concise and better at commonsense reasoning. S1 models showed higher confidence with lower entropy, whereas S2 models exhibited more uncertainty. The authors also trained hybrid models with varying S1/S2 ratios and developed a training-free approach that dynamically selects between S1 and S2 outputs based on confidence metrics, improving performance on most benchmarks.

**Strengths:**

- The methodology is well-grounded in cognitive science theory and implemented through a carefully controlled dataset with rigorous safeguards against superficial pattern learning.
- The empirical evaluation comprehensively demonstrates meaningful task-dependent performance patterns, while mechanistic analysis provides deep insights into how the models differ behaviorally and confirms that reasoning styles form a continuous spectrum rather than discrete categories.

**Weaknesses:**

-  All experiments use Llama-3-8B and Mistral-7B. No evidence for generalization to smaller models (1-3B) with potentially insufficient capabilities, or larger models (70B+) with different emergent abilities.
- The authors normalize S1/S2 response lengths to prevent superficial learning. However, in authentic S1/S2 reasoning, length differences are intrinsic—S1 responses should naturally be shorter. Equalizing lengths during training may teach models an unnatural constraint.

**Questions:**

- Can you test on one larger model to validate that S1/S2 distinctions persist at scale? Do larger models naturally exhibit more S2-like behavior?
- You normalized S1/S2 response lengths to avoid superficial cues, but S2 models still produced longer outputs at inference. I wonder if an ablation training on un-normalized data may quantify how much "overthinking" behavior is inherent to S2 alignment versus an artifact of training on longer examples.

---

> ### Author Response · Authors · 2025-11-16
>
> We sincerely thank the Reviewer for acknowledging the cognitive science grounding of our work, the rigor of our data construction, and the comprehensive empirical and mechanistic analyses. Below, we address the concerns raised.
>
> Questions and Weaknesses Response:
>
> 1. We did as the Reviewer suggested. We conducted additional experiments on both a smaller model (Llama-3.2-3B) and a larger model (Llama-3-70B), using the same exact setup. The results are shown in the Tables below. Across both model sizes, we observe the same qualitative patterns reported in the main paper: S2-aligned models excel in arithmetic and symbolic reasoning, while S1-aligned models perform better in commonsense reasoning tasks. This confirms that the S1 and S2 distinction persists at scale and is not an artifact of mid-sized models. Overall, based on all of our experiments, this trend is architecture-agnostic (Llama vs. Mistral), scale-robust (3B, 8B, 70B), and algorithm-robust (DPO vs. SimPO). We will include these results in the updated manuscript once the dynamic model runs have been completed, and will let you know.
>
> Llama-70B
> | Benchmark | base | S1 (DPO) | S2 (DPO) | S1 (SIMPO) | S2 (SIMPO) |
> | --- | --- | --- | --- | --- | --- |
> | MultiArith | 98.21 | 97.83 | 98.99 | 97.5 | 98.61 |
> | GSM8K | 94.68 | 93.91 | 96.74 | 94.28 | 95.37 |
> | AddSub | 85.93 | 82.47 | 89.68 | 81.94 | 91.45 |
> | AQuA | 50.87 | 48.82 | 51.06 | 49.69 | 55.53 |
> | SingleEq | 91.32 | 85.59 | 94.83 | 90.23 | 95.12 |
> | SVAMP | 83.7 | 80.3 | 86 | 83.1 | 86 |
> | Coin | 95 | 93.6 | 94.8 | 92.8 | 95 |
> | Letter | 88.2 | 87.8 | 90.2 | 87.6 | 88 |
> | CSQA | 73.56 | 75.32 | 69.62 | 73.87 | 73.49 |
> | Strategy | 69.06 | 70.87 | 61.39 | 69.62 | 67.73 |
> | PIQA | 85.77 | 86.19 | 85.46 | 85.85 | 83.94 |
> | SIQA | 74.89 | 75.64 | 72.06 | 75.32 | 73.68 |
> | COM2SENSE | 80.49 | 81.09 | 76.42 | 81.46 | 79.99 |
>
>
> Llama-3B
> | Benchmark | base | S1 (DPO) | S2 (DPO) | S1 (SIMPO) | S2 (SIMPO) |
> | --- | --- | --- | --- | --- | --- |
> | MultiArith | 75.32 | 74.91 | 75.88 | 75.16 | 75.58 |
> | GSM8K | 68.76 | 67.36 | 69.73 | 68.03 | 69.43 |
> | AddSub | 70.48 | 68.29 | 74.45 | 68.11 | 73.96 |
> | AQuA | 30.51 | 29.84 | 32.01 | 30.27 | 32.84 |
> | SingleEq | 66.34 | 64.97 | 67.44 | 62.66 | 68.29 |
> | SVAMP | 53.9 | 51.5 | 54.3 | 51.3 | 54.1 |
> | Coin | 78 | 75.8 | 76.8 | 77.2 | 78 |
> | Letter | 66.4 | 63.8 | 66.4 | 64.2 | 66 |
> | CSQA | 62.39 | 61.73 | 59.27 | 62.34 | 59.97 |
> | Strategy | 69.06 | 69.48 | 67.95 | 70.18 | 66.78 |
> | PIQA | 73.15 | 74.61 | 71.23 | 74.53 | 72.42 |
> | SIQA | 45.82 | 45.82 | 44.96 | 46.46 | 44.71 |
> | COM2SENSE | 59.97 | 60.08 | 58.34 | 59.97 | 58.89 |
>
>
> 2. We thank the Reviewer for suggesting the ablation study. We conducted additional analyses by training our models without length normalization and we have added the results in section 5.2 and appendix M. We found that S2 responses are significantly longer than S1 responses for both DPO (t(8836) = 71.831, p < .001) and SIMPO (SIMPO t(8586) = 15.227, p < .001) when the text is not normalized. This is consistent with our Section 5.2 results, where S2-aligned models also produced longer outputs when trained on length-normalized data (DPO: t(8836)=57.14, p<.001; SIMPO: t(8586)=9.833, p<.001). In both settings, the S2 models generate more tokens compared to their S1 counterparts. These suggest that the “overthinking” behavior is inherent to the S2 reasoning style. As noted in Lines 210-212, prior work has shown that alignment methods can rely on superficial cues such as output length, which may bias models toward longer responses even without genuine reasoning improvement [1]. Therefore, our length normalization was intended to remove this cue and ensure the models learned the underlying reasoning processes rather than only reproducing longer outputs. Moreover, as described in Lines 203–206 and Appendix H, our dataset is process-oriented, and experts verified that normalization only trimmed redundant elaboration while preserving the S2 reasoning structure.
>
> [1] Singhal, Prasann, et al. "A long way to go: Investigating length correlations in rlhf." arXiv preprint arXiv:2310.03716 (2023).

---

> > ### Author Response · Authors · 2025-11-21
> > **3B and 70B added to the paper!**
> >
> > Following up on our previous response, we have completed the experiments on model scaling as suggested. We have updated the revision to include results for the S1, S2, and dynamic models (across both alignment algorithms) on Llama-3 3B, 8B, and 70B.
> >
> > For clarity and organization, Table 1 now presents the Llama-3 results, while the Mistral results have been moved to Table 6 in Appendix Q. We find that the performance improvements are robust and consistent with the original 8B results, demonstrating scalability across model sizes.

---

> ### Comment · Reviewer_6SYb · 2025-11-23
>
> I appreciate the authors for the additional results. I have raised my soundness and contribution evaluation scores accordingly.

---

### Comment · Area_Chair_hokU · 2025-11-24
**Author Responses Are Ready - Please Review & Provide Feedback**

Dear Reviewers,

Thank you once again for your essential contributions to the review process. The authors have submitted their responses to your initial reviews.

I kindly ask you to carefully review the authors' responses for the papers you are handling. Your timely assessment of how the authors have addressed your original concerns is a critical step in reaching a final decision.

Please provide your feedback and any necessary updates to your reviews as soon as possible to ensure we can meet our tight schedule for the discussion phase.

Your prompt attention to this matter is highly appreciated.

Best regards,

Area Chair

---

### Author Response · Authors · 2025-11-25
**Summary of Revisions**

Hi,

We would like to thank all the reviewers for their thoughtful and constructive feedback.

A brief note on the updates in our revision: we added additional models of various sizes, improved the overall writing, and included numerical results for the dynamic models in Table 1. In addition, as part of the evaluation on the new dataset requested by Reviewer AWgf, some numerical values in our analyses have changed slightly; however, all original conclusions remain robust.

Best regards,

---

### Author Response · Authors · 2025-12-03
**Summary Response (1/2)**

Dear Area Chair,

We would like to express our sincere appreciation for the time and care invested throughout the review process. We are also grateful to all reviewers for their positive and constructive feedback and for the thoughtful discussions during the rebuttal, which helped us strengthen the paper. Below, we summarize each Reviewer’s concerns, the corresponding revisions, clarifications, or additional experiments we performed. Our goal is to make it easy to see how each issue was resolved without affecting the paper’s main findings.

| Reviewer | Question and Weakness | Author Response | Original Score | Changed Score |
|----------------------|----------------------|----------------------|----------------------|----------------------|
| 6SYb | 1. Lack of generalization to smaller (1-3B) or larger (70B) models; asked for validation at other model scales | 1. Conduct additional experiments on Llama-3 3B and Llama-3 70B and report results for 3B, 8B, 70B; trends hold across scales and architectures. (added to Table 1 and moved Mistral to Table 6 in Appendix Q) | 6 | 6 (increased soundness and contribution scores from 3 to 4)|
| | 2. Concern that length normalization is unnatural; requested ablation. | 2. Conduct an ablation study by training without length normalization and report that S2 responses remain significantly longer, concluding that longer outputs are inherent to S2 alignment; kept normalization to avoid superficial cues. (added to Appendix M) | | |
| AWgf | 1. Concern that length-matching may alter natural S1 brevity; asked for training without normalization. | 1. Conduct an ablation study by training without length normalization, and report that S2 responses remain significantly longer. (added to Appendix M) | 4 | 4 (no answer) |
| | 2. Noted that the entropy-based dynamic approach is effectively an ensemble (inference cost). | 2. Acknowledged the dynamic method is an ensemble and noted increased inference cost in limitation (Lines 869-872); added a clear statement about the increased inference cost and single model distillation as an important direction for future work to further clarify this nuance (added to Conclusion; Lines 479-481). | | |
| | 3. Math benchmarks may be too simple; asked for evaluation on deeper multi-step math and whether S2 advantage holds on complex problems, and becomes more distinct. | 3. Expanded evaluation to include AGIEval, a more multi‑step math benchmark (avg. 348.1 tokens per response vs. 168.2 for our other math benchmarks used in our experiments); the S2‑aligned model outperforms both S1 and base models, and the distinction in reasoning depth is more pronounced since the average output length gap widens from 304.6 (S1) to 391.7 (S2) tokens. | | |
| | 4. Concern about out-of-distribution transfer and whether alignment effects generalize beyond the alignment dataset. | 4. Clarified alignment was performed only on the custom reasoning-style dataset, and all downstream benchmarks were evaluated zero-shot; therefore, reported results already reflect OOD transfer | | |
| B6EU | 1, 2. The dynamic models' results were missing quantitative results in the main text. The methodology was primarily illustrated via GSM8K figures; requested adding rows for dynamic models in Table 1. | 1, 2. Added quantitative results for the dynamic model to Table 1. | 2 | 2 (no answer) |
| | 3. Concern that parameter w was tuned per dataset (unfair evaluation). | 3. Clarified that the dynamic model uses a fixed w=0.4 across all benchmarks; added performance curves to Appendix U to show that the optimal region consistently centers around 0.4 regardless of the dataset, and the choice of w depends on the intrinsic behavior of the S1 and S2 models. | | |
| | 4. The models used seem outdated; Suggest evaluating them on more recent models to demonstrate state-of-the-art performance. | 4. Emphasized the paper’s primary contribution is analysis of S1 vs S2 behaviors, not state-of-the-art performance; however, conduct additional experiments on Llama‑3 3B and 70B (and Mistral in Appendix Q) showing the S1/S2 phenomena are consistent across architectures and scales.  | | |
| | 5. Writing is repetitive; the main text lacks crucial quantitative examples while the appendix holds key experiments. | 5. Improved presentation: added quantitative dynamic results to the main text, included additional model‑scale tables (3B, 8B, 70B), and asked for guidance if any other tables or figures in the appendix are important to move to the main text, then if the page limit exceeds, we are prepared to shorten the conclusion. | | |

---

> ### Author Response · Authors · 2025-12-03
> **Summary Response (2/2)**
>
> | Reviewer | Question and Weakness | Author Response | Original Score | Changed Score |
> |----------------------|----------------------|----------------------|----------------------|----------------------|
> | 2qiX | 1. Possible contradiction between mentioning prompt-based S1/S2 elicitation and using alignment training in experiments. | 1. Clarified that prompting was only a motivational context; the method *entirely* relies on alignment training. Updated wording in Sec. 3.1 removes the ambiguity. | 6 | 6 (answered that there is no further question and kept their score) |
> | | 2. The reliability metric seemed insufficiently justified (entropy + variance). | 2. Rewrote the explanation of the reliability metric to show that entropy and entropy-variance act as complementary token-level stability signals, consistent with prior work. No methodological change needed. | | |
> | | 3. The dynamic model requires both the S1 and S2 models, raising concerns about efficiency. | 3. Acknowledged inference-cost limitation in Limitations and Conclusion; noted this does not affect core findings and can be addressed with future single-model distillation. | | |
> | | 4. Questions: (a) effect of few-shot CoT, (b) examples of S1/S2 with similar length, (c) whether shortening S2 harms reasoning quality. | 4. Completed all requested checks: added few-shot CoT experiments (conclusions unchanged), provided matched-length S1/S2 examples (Appendix H), and added ablations confirming S2 length is intrinsic and shortening removes redundancy only. None of these issues affects the main results. | | |
> | rtgL | 1. Asked why S2 answer length increases after the two‑stage prompt; requested a concrete example and a table showing accuracy and token counts for GSM8K, Coin, and CSQA to illustrate S1 and S2 models' reasoning characteristics differences. | 1. Provided statistical analysis showing marginal differences in Stage 1 and a large, significant difference in Stage 2, and further clarified our findings regarding the reason behind this observation; provided a concrete Coin example contrasting S1 vs. S2 Stage 1/Stage 2 outputs to better illustrate the reasoning differences; added a table with S1/S2 accuracy and response lengths per the reviewer’s request (added Tables 8 and 9 in Appendix S). | 6 | 8 (increased their overall score) |
> | | 2. Asked about the parallels between the findings of our paper regarding the differences in reasoning between S1 and S2 models and the differences in reasoning (e.g., Aha moment) between reasoning models (e.g., DeepSeek R1) and non-reasoning models. | 2. Explained the differences in the Aha-moment characteristic of reasoning models as well as the reasoning characteristics of our S2 models, how their long reasoning chains are similar to each other, and further clarified the differences between them. | | |

---

### Meta-Review · Area_Chair_c3zf · 2026-01-06

**Summary:**

This paper questions on whether System 2 (S2) reasoning is uniformly better than System 1 (S1) in LLMs. The authors curate a dataset of 2,000 questions with paired responses for both reasoning styles. Using RLHF algorithms like DPO and SimPO, they align models to either S1 or S2 reasoning. The result identifies that S2 models excel in arithmetic tasks but perform worse than S1 models on commonsense tasks. Additionally, the paper proposes a dynamic model that arbitrates between the two styles using entropy and the variance of entropy in a training-free manner. This method achieves improved performance across most benchmarks in both arithmetic and commonsense tasks.

**Reviewer Concerns:**

Addressed Concerns:
- Generalization to Model Scales: Reviewers raised concerns that initial experiments are conducted on 7B and 8B models, which might not generalize to smaller or larger models. The authors addressed this by conducting additional experiments on 3B and 70B models, the results of which indicating that the conclusions hold consistently across scales.
- Benchmark Difficulty: The authors incorporated AGIEval, which involves complex multi-step reasoning, in response to reviewers’ comment on using too simple mathematical benchmarks. The results in AGIEval remains consistent with the claims in the paper.
- Quantitative Results for Dynamic Models: The authors revised their Table 1 to include quantitative results for the dynamic model, showing that it outperforms baselines on most of the benchmarks.
- Hyperparameter Tuning: Reviewer B6EU expressed concerns on whether the authors have tuned the hyperparameter w for each of the datasets, resulting in unfair comparison. The authors clarified that they have used a consistent $w=0.4$ for all datasets.

Outstanding Concerns:
- Inference Cost of the Dynamic Models: Reviewers noted that the proposed dynamic model is an ensemble rather than a switch that requires running both S1 and S2 models, thus increasing the inference cost. The authors acknowledged this limitation and suggested future distillation.
- Length Normalization: Some reviewers worried that normalizing response lengths during training might introduce unnatural constraints. The authors conducted another group of experiments without length normalization, confirming that S2-aligned models generate longer responses and tend to overthink. However, these results are not sufficient to illustrate that normalizing response lengths introduces no spurious constraints.
- Outdated Backbones: Reviewer B6EU pointed out that the backbones are somewhat outdated. The GPT-4o model might not have the capability to authentically distinguish between S1 and S2 reasoning. Using more recent models that feature explicit S1 and S2 reasoning modes helps the authors’ claims to become more pursuasive.

**Reviewer Scores:**

- Reviewer 6SYb (Score: 6): Maintained 6.
- Reviewer AWgf (Score: 4): Would likely remain at 4 as the fundamental critique regarding length-matching and the inference cost remains unaddressed..
- Reviewer B6EU (Score: 2): Would likely remain at 2 as the fundamental critique regarding outdated backbones remains unaddressed.
- Reviewer 2qiX (Score: 6): Maintained 6.
- Reviewer rtgL (Score: 6): Improves to 8.

---

### Decision · Program_Chairs · 2026-01-26

Reject